# Nonlinear Vibration Characteristics of Virtual Mass Systems for Wind Turbine Blade Fatigue Test

Aiguo Zhou[1], Jinlei Shi[1], Tao Dong[1], Yi Ma[1], Zhenhui Weng[2]

[1]School of Mechanical Engineering, Tongji University, Shanghai 200082

[2]Aeolon Technology Co., Ltd, Shanghai 200120, China

*Correspondence to*: *Jinlei Shi* (*shijinlei1430@163.com*)

**Abstract.** The biaxial fatigue test of wind turbine blades is helpful to shorten the test time and is more suitable for the actual operating conditions. Adding tuning masses to the blade is a common method for blade uniaxial test at present, and its purpose is to adjust the load distribution in one direction of the blade. However, the tuning masses on the blade will affect the load distribution in the direction of the blade flap-wise and edge-wise at the same time in the biaxial test, so the concept of "virtual masses" is proposed to realize the decoupling of the load distribution in the biaxial test. Due to the limitation of the size of the virtual masses mechanism and the complex motion trajectory of the blade, the actual inertial effect provided by the virtual masses is different from the ideal situation, which will affect the resonance characteristics of the test system and the load distribution of the blade. Therefore, in order to evaluate the effect of the nonlinear effect introduced by the virtual masses on the resonance characteristics of the test system and the blade load distribution, the equivalent dynamic model of the bladed virtual mass test system was established by using the Lagrange method. Then, the nonlinear effects of blade amplitude and virtual mass installation parameters on the test system are obtained by numerical method. Then, based on the nonlinear vibration theory, the approximate nonlinear amplitude-frequency characteristics of the test system are obtained, that is, the resonance frequency of the test system will decrease with the increase of the blade amplitude. Through the simulation analysis of two 80m+ blades, the applicability of the theoretical method is verified. It can be seen from the simulation results of the simulated uniaxial test that the larger the amplitude of the blade and the shorter the connection rod will reduce the resonance frequency of the test system. When the vibration amplitude at the excitation point is the same, a lower resonance frequency results in a smaller load distribution level, that is, the area which is actually fully tested will be reduced. In the biaxial simulation test, the resonance frequency of the test system will be further reduced because the virtual masses will be affected by the coupled motion in both directions at the same time. Besides, the introduction of an external mechanism of the virtual mass will also cause deformation of the envelope of the blade biaxial trajectory, which will further affect the load distribution of the blade. This work explores the nonlinear influence of virtual masses on the actual fatigue test, provides the corresponding theoretical basis and reference for the test organization to adjust the tuning masses scheme in advance to adjust the load distribution and select the excitation equipment.

## 1 Introduction

As an important component of wind turbine, the cost of blades accounts for 20% of the overall machine, so the lifetime of blades is the premise to ensure safe and stable operation of the wind turbine (Zhang et al., 2015; Liao et al., 2016). To verify the reliability of the blade under the actual operating field, the International Electrotechnical Commission (IEC) points out that the full-scale fatigue test of rotor blades is needed to be performed (IEC, 2014). In the actual blade fatigue test, two separate oscillation tests with over one million damage-equivalent loads cycles are usually performed.

The fatigue test requires that the load in the area of interest along the blade span-wise direction matches or exceeds the design value, while keeping the exceedance as small as possible in order to avoid unrealistic failures (DNV GL AS, 2015). To satisfy the above requirements, additional masses are usually attached to the blade to tune the test load distribution which needs to be optimized by determining the optimal masses distribution.

To save testing time and to emulate the comprehensive damage along the circumference of the blade, several institutions

began to study and design biaxial fatigue test (White et al., 2004; Greaves et al., 2012; Snowberg et al., 2014; Hughes et al., 1999; Liao et al., 2014;), namely to excites the blade in both directions simultaneously. In the previous biaxial resonance test, a reasonable load distribution (in both directions) will be obtained by optimizing the position and tuning masses installed on the blade. However, the tuning masses installed on the blade will affect the vibration characteristics (mode shape and frequency) in both flap-wise and edge-wise directions, which brings difficulty to the biaxial load match optimization, and there may be excessive overload in a certain area of the blade when choosing a compromise.

To simplify load match, the extra mechanism makes the tuning masses only act in one vibration direction (called virtual masses). The purpose of the virtual masses is to decouple the biaxial load, so that the biaxial load match is equivalent to the combination of the load match of two single axis test. Post et al. (2016) firstly proposed the concept of virtual masses to tune both natural frequencies independently in the two directions, and to eliminate the coupling phenomenon of test bending moments during biaxial test. Melcher et al. (2020a、2020b) used elastic elements to adjust blade stiffness, and optimized biaxial fatigue test parameters based on virtual masses and elastic elements. Zhang et al. (2020) and Lu et al. (2022) carried out research on biaxial load matching and design using virtual masses. The virtual masses used for mass decoupling is ideally regarded as translational motion and the push rod between the virtual mass and the blade is always in line with the main vibration in the above work, which is difficult to apply to the actual test field. Because a larger and stronger platform is needed to keep virtual mass translate in the edge-wise direction, which is difficult to achieve in a limited test space. In the biaxial test, the platform may interfere with the push rod, especially when the blade has a large amplitude in the flap-wise direction. Therefore, IWES conducted further research, designed a device to convert virtual masses from translation to rotation, and applied it to the biaxial fatigue test which has a frequency ratio of 1:1 (Melcher et al., 2020c). Further, the feasibility of the biaxial decoupling test of the bending moment was verified by the comparison of simulation and experiment results (Melcher et al., 2020c; Castro et al.,2021; Falko et al., 2020). In fact, in the view of the motion characteristics, the inertia force generated by rotating virtual masses is different from that generated by translational virtual masses. Taking a uniaxial test as an example, the translational virtual masses move synchronously with the blade, which behaves like a mass acting in just one direction from a numerical standpoint. The translational virtual masses have the same motion characteristics as the additional tuning masses. Therefore, although the virtual mass is not on the blade, the inertia force generated by it and the inertia force generated by the additional tuning masses are in the same direction and magnitude. The rotating virtual masses are limited by the constraints of the seesaw, and its motion path is the rotating motion around the center of the seesaw. Therefore, the direction and magnitude of the inertial force generated by the rotation of the virtual mass will change, and it is not equivalent to the translational virtual masses. However, changes in the inertia force provided by the virtual masses will cause changes in the characteristics of the system, which may further cause changes in the blade load distribution, and may put forward higher requirements for vibration excitation equipment.

To reveal the vibration mechanism of the blade-virtual masses test system and provide a more rigorous theoretical basis for the biaxial load matching theory of the blade. In this paper, a theoretical model of blade-virtual masses uniaxial test system is established. The specific nonlinear impact of single parameter related to virtual masses on the characteristics of the test system can be obtained intuitively through the uniaxial model. Then, two blades over 80m were simulated in ADAMS. Uniaxial simulation was used to verify the applicability of the theoretical model, including the nonlinear amplitude-frequency characteristics of the system and the effects of virtual mass installation parameters (such as seesaw length) on the load distribution of the blade. Biaxial simulation is used to analyze the nonlinear effect of virtual mass on the system under the simultaneous action of many factors. This work will be used in the future research to adopt reasonable control strategy and adjust the counterweight scheme in advance to achieve the target damage of the blade.

## 2 Blade-virtual masses equivalent dynamic model

The tuning masses can change the modal characteristics of the testing system to adjust the test load distribution of the blade, which is essentially bending moment caused by the inertia force brought by the reciprocating motion of the self-weight and tuning masses. In the common uniaxial fatigue test system, the tuning masses are directly attached to the blade, as shown in Fig. 1 (a). When the tuning masses are determined, the modal characteristics of the testing system are basically determined. This means that, without considering the air damping, the resonance frequency of the system remains unchanged.

In the biaxial fatigue test, the tuning masses decouple the biaxial load by seesaw, and the tuning masses are called virtual masses, as shown in Fig. 1 (b). The inertia force generated by the virtual masses mainly acts in the edge-wise direction in Fig. 1 (b). The mechanism for mounting the virtual masses consists of a push rod and a seesaw. The blade fixture, push rod, and seesaw are connected through a universal joint, and the seesaw can rotate around the center position. Tuning masses are located at both ends of the seesaw to offset each other's gravity. After the exciting force is applied to the blade, the tuning masses move with the blade and rotate around the center of the seesaw to provide the inertia force for the blade through the push rod. However, due to the motion characteristics of the virtual masses mechanism, the motion of the virtual masses cannot be perfectly synchronized with the blade motion. Therefore, the inertia force generated by the rotation of the virtual masses differs from the inertia force generated by the traditional tuning masses. To evaluate the specific impact of single parameter related to virtual masses on the test system, it is necessary to establish the corresponding uniaxial theoretical model for analysis from the perspective of control variable method.

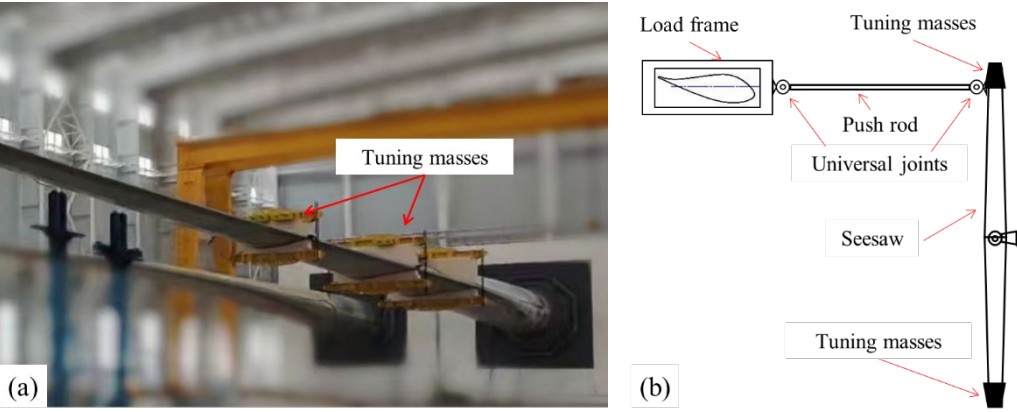

**Figure 1: Masses match of blade fatigue test: (a) traditional tuning masses setup (b) virtual masses setup.**

### 2.1 The comparison on the amplitude of inertia force

The uniaxial test is taken as an example to illustrate the difference between virtual masses translation and rotation, as shown in Fig.2. The inertial force generated by rotating virtual masses of the blade at the maximum amplitude can be analyzed, as shown in Fig.3. The relationship of the motion between virtual masses and blade can be obtained:

$$\begin{cases} \boldsymbol{v}_m = \boldsymbol{v}_M + \boldsymbol{v}_{mM} \\ \boldsymbol{a}_m = \boldsymbol{a}_m^n + \boldsymbol{a}_m^\tau = \boldsymbol{a}_M + \boldsymbol{a}_{mM}^n + \boldsymbol{a}_{mM}^\tau \end{cases} \tag{1}$$

Where: $\boldsymbol{v}_m$ - velocity of virtual masses; $\boldsymbol{v}_M$ - velocity of blade equivalent mass; $\boldsymbol{v}_{mM}$ - relative velocity; $\boldsymbol{a}_{mM}^n$ - relative normal acceleration; $\boldsymbol{a}_m$ - the acceleration of the virtual masses; $\boldsymbol{a}_{mM}^\tau$ - relative tangential acceleration; $\boldsymbol{a}_m^n$ - normal acceleration; $\boldsymbol{a}_m^\tau$ - tangential acceleration.

The blade at the maximum amplitude satisfies: $\boldsymbol{v}_M = 0$; $\boldsymbol{v}_{mM} = 0$; $\boldsymbol{a}_{mM}^n = 0$; $\boldsymbol{a}_m^n = 0$.

The angular acceleration of the virtual mass at the maximum amplitude of the blade can be obtained:

$$|\alpha_m| = \frac{\omega^2 Y \cos(\beta_0)}{R \cos(\theta_0 - \beta_0)} \tag{2}$$

Where: $\theta$ - Rotation angle of the seesaw at the maximum amplitude of the blade; $\beta_0$ - Angle between the push rod and the main vibration direction at the maximum amplitude of the blade; $\alpha_m$ - Angular acceleration of the virtual mass at the maximum amplitude of the blade.

According to Eqs. (1) and Eqs. (2), the rotating inertia force $F_R$ generated by the rotating virtual mass at the maximum
amplitude of the blade can be obtained:
$F_R = \dfrac{m\omega^2 Y \cos(\beta_0)}{\cos(\theta_0 - \beta_0)}$          (3)
The inertia force $F_{rot}$ transmitted to the main vibration direction of the blade through the push rod can be obtained:
$F_{rot} = \dfrac{F_R \cos(\beta_0)}{\cos(\theta_0 - \beta_0)} = \dfrac{m\omega^2 Y \cos^2(\beta_0)}{\cos^2(\theta_0 - \beta_0)}$       (4)
The translational virtual masses are consistent with the motion state of the blade, so the inertial force generated by the
translational virtual masses can be obtained based on Eqs. (4):
$F_{tra} = m\omega^2 Y$              (5)
According to Eqs. (4) and Eqs. (5), there are differences in the inertial forces acting on the blades by the two setups, which
are mainly caused by the difference in the movement trajectory of masses.

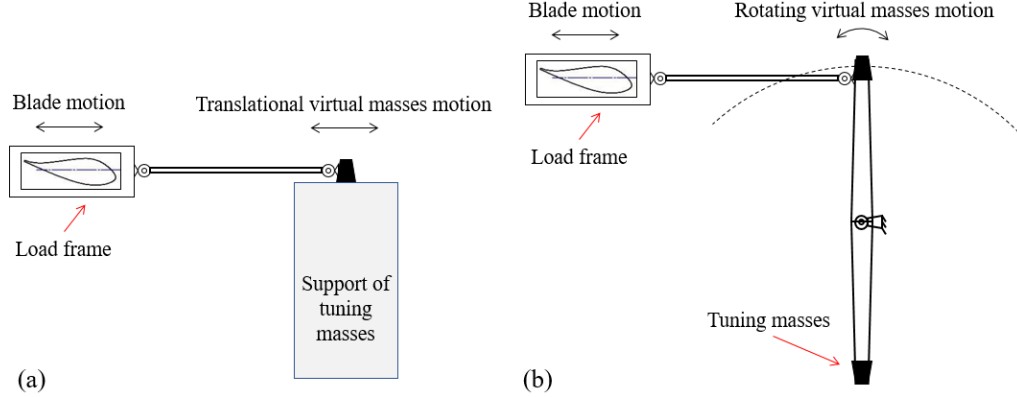


**Figure 2: The comparison on inertia force: (a) Translational virtual masses setup (b) Rotating virtual masses setup.**
**2.2 Model using the Lagrange method**
In fact, there will be inertial force coupling in the actual biaxial testing process (virtual masses translation or rotation), which
will cause multiple factors to work together and make it difficult to analyze the system characteristics quantitatively by
theoretical method. Therefore, it is desirable to choose the uniaxial test to analyze the nonlinear influence introduced by the
virtual masses, which does not mean that the biaxial test can be regarded as the linear superposition of the uniaxial test.
Essentially, the load distribution in the main vibration direction of the blade is adjusted by the component of the inertia force
transmitted by the push rod in this direction. Because of the angle between the push rod direction and the vibration direction,
blade displacement is not in line with the push rod. To more intuitively analyze the impact of virtual masses on the blade test
system, the mass of the push rod and the seesaw are ignored in modeling according to the control variable method, and
only their geometric dimensions are considered. Take the example of blade edge-wise direction test, the blade model is
simplified as shown in Fig. 3. Moreover, the inertial force of the virtual masses also affects the flap-wise direction of the blade.
However, since the frequency of the inertial force is close to the first order modal frequency in edge-wise direction, the
perturbation to the flap-wise direction is relatively small. Therefore, only the influence of virtual masses on the vibration
characteristics in the main testing direction needs to be considered during the uniaxial test. Section 2.1 only analyzes the
difference of inertial force amplitude in Fig. 2 and this section set up a uniaxial theoretical model to evaluate the effect of
virtual masses rotation on the vibration characteristics of the test system. In this paper, the Lagrange method is used to analyze
the uniaxial model (Liu et al., 2019). The initial state of the test system is assumed when the blade is stationary, the push rod
is horizontal and the seesaw is vertical.

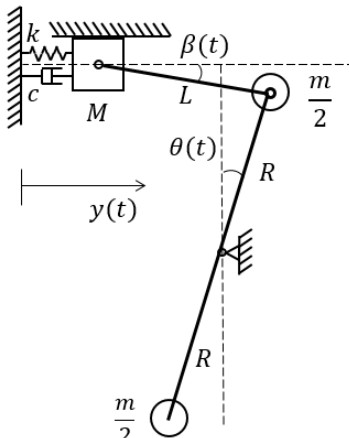


**Figure 3: Virtual masses setup for blade fatigue test.**
$$\frac{d}{dt}\left(\frac{\partial T}{\partial \dot{q}_j}\right) - \frac{\partial T}{\partial q_j} + \frac{\partial V}{\partial q_j} + \frac{\partial D}{\partial \dot{q}_j} = Q_j, j = 1,2,\cdots,n \tag{6}$$
Where: $T$ - kinetic energy; $V$ - potential energy; $D$ - dissipated energy; $q_j$ - generalized coordinate; $\dot{q}_j$ - generalized
velocity; $Q_j$ - generalized force.
By selecting the generalized coordinate $q = y$, and based on the motion relationship in Fig. 3, the displacement and
velocity relationships of the test system can be obtained:
$$\begin{cases} y + L\cos\beta - R\sin\theta = L \\ L\sin\beta + R\cos\theta = R \end{cases} \tag{7}$$
$$\begin{cases} \dot{y} - L\dot{\beta}\sin\beta - R\dot{\theta}\cos\theta = 0 \\ L\dot{\beta}\cos\beta - R\theta\sin\theta = 0 \end{cases} \tag{8}$$
$T$, $V$ and $D$ can be calculated as
$$T = \frac{1}{2}M\dot{y}^2 + \frac{1}{2}mR^2\dot{\theta}^2 = \frac{1}{2}M\dot{y}^2 + \frac{1}{2}m\dot{y}^2\frac{\cos^2\beta}{\cos^2(\theta-\beta)} \tag{9}$$
$$V = \frac{1}{2}ky^2 \tag{10}$$
$$D = \frac{1}{2}c\dot{y}^2 \tag{11}$$
Where: $L$ - the length of the push rod; $R$ - the radius of the seesaw; $\beta$ - the angle between the push rod and the horizontal
direction; $\theta$ - the angle between the seesaw and the vertical direction; $M$ - blade equivalent mass; $m$ – virtual masses; $k$ -
blade equivalent stiffness; $c$ - blade equivalent damping.
According to Eqs. (7) and Eqs. (8), the relevant terms in Eqs. (6) are obtained as
$$\begin{cases} \frac{d}{dt}\left(\frac{\partial T}{\partial \dot{y}}\right) = M\ddot{y} + m\ddot{y}\frac{\cos^2\beta}{\cos^2(\theta-\beta)} + m\dot{y}\frac{d}{dt}\left[\frac{\cos^2\beta}{\cos^2(\theta-\beta)}\right] \\ \frac{\partial T}{\partial y} = \frac{1}{2}m\dot{y}^2\frac{\partial}{\partial y}\left[\frac{\cos^2\beta}{\cos^2(\theta-\beta)}\right] \\ \frac{\partial V}{\partial y} = ky \\ \frac{\partial \Psi}{\partial \dot{y}} = c\dot{y} \\ Q(t) = F(t) \end{cases} \tag{12}$$
Then, the dynamic differential equation of test system is obtained as
$\left\{M + m\dfrac{\cos^2\beta}{\cos^2(\theta-\beta)}\right\}\ddot{y} + c\dot{y} + ky + \dfrac{m\dot{y}^2\cos\beta}{\cos^4(\theta-\beta)}\left[\dfrac{\cos^2\beta\sin(\theta-\beta)}{R} - \dfrac{\sin^2\theta}{L}\right] = F(t)$    (13)
By comparison with Eqs. (4), it can be seen that the inertial force terms of two equations are same at the maximum
amplitude of the blade.
Where:
$\sin\theta = \dfrac{L+y}{R} - \dfrac{L\left(R\sqrt{-(y^2+2Ly-2LR)(y^2+2Ly+2LR)}+y^3+2L^3+4L^2y+3Ly^2\right)}{2R(L^3+2L^2y+LR^2+Ly^2)}$
$\cos\theta = \dfrac{L(L+y)[R\sqrt{-(y^2+2Ly-2LR)(y^2+2Ly+2LR)}+2L^3+y^3+3Ly^2+4L^2y]}{2R^2(L^3+2L^2y+LR^2+Ly^2)} - \dfrac{2L^2+2Ly-2R^2+y^2}{2R^2}$
$\sin\beta = \dfrac{2L^2+2Ly+y^2}{2LR} - \dfrac{(L+y)[R\sqrt{-(y^2+2Ly-2LR)(y^2+2Ly+2LR)}+2L^3+y^3+3Ly^2+4L^2y]}{2R(L^3+2L^2y+LR^2+Ly^2)}$
$\cos\beta = \dfrac{R\sqrt{-(y^2+2Ly-2LR)(y^2+2Ly+2LR)}+2L^3+y^3+3Ly^2+4L^2y}{2(L^3+2L^2y+LR^2+Ly^2)}$
According to Eqs. (13), it can be seen that rotation of virtual masses introduces nonlinear terms to the test system, and both
the angle $\theta$ and $\beta$ are nonlinear functions of the blade response $y$. Due to the complexity of the dynamic equation, it is
difficult to obtain the corresponding analytical expression. Therefore, the numerical analysis methods are used to solve the
equation. A numerical simulation model based on the differential equation of the system motion is established in MATLAB
SIMULINK, and the corresponding resonance frequency of the equivalent system can be obtained by setting different initial
displacements. By modifying the value of the different parameter, the influence of the parameter change on the resonance
frequency of the test system can be obtained. As mentioned previously, the nonlinear factors that affect the characteristics of
the test system mainly come from installation parameters (pushrod length and seesaw radius) and blade response. The design
length of the push rod generally remains unchanged due to space limitations at the test site. However, the seesaw radius offers
greater design flexibility. Thus, the primary focus is on evaluating the impact of the seesaw radius $R$ and blade response $y$
on the vibration characteristics of the blade. To illustrate this, the equivalent parameters of 80m blade are brought into the
differential equation and numerically analyzed, and the influence of blade amplitude on the resonance frequency of the test
system is investigated, as demonstrated in Fig. 4.
Figure 4 (a) shows that the resonance frequency of the test system decreases nonlinearly with an increase in blade
amplitude and virtual masses $m$ further determines the rate of decrease in resonance frequency. The equivalent stiffness $k$
has the ability to alter the natural frequency of the test system. However, it can be seen from Fig. 4 (b) that $k$ cannot change
the rate of decrease in resonance frequency with other parameters unchanged, which indicates that the equivalent stiffness is
not a nonlinear factor affecting the vibration characteristics of the testing system. Fig. 4 (c) shows that the increase of $M$ will
delay the decline rate of the natural frequency of the system, because the proportion of the virtual masses in the inertia force
term decreases. It can be seen from Fig. 4 (d) that the radius of the seesaw will also affect the nonlinear amplitude-frequency
characteristics of the test system and the rate of decrease in resonance frequency.

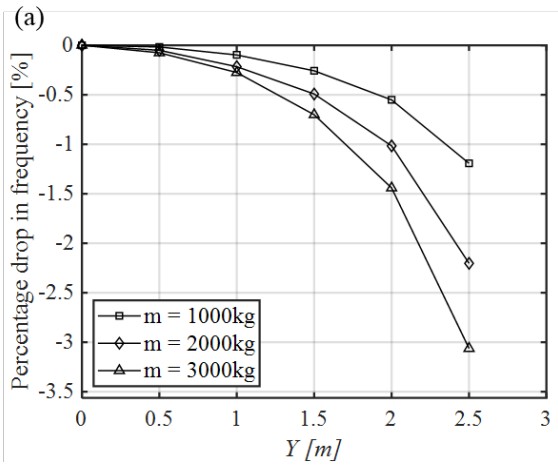
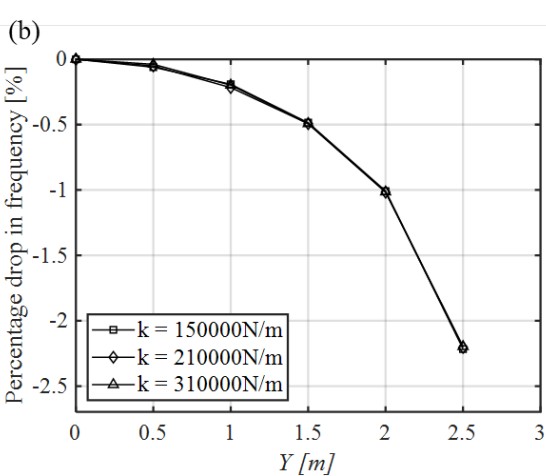


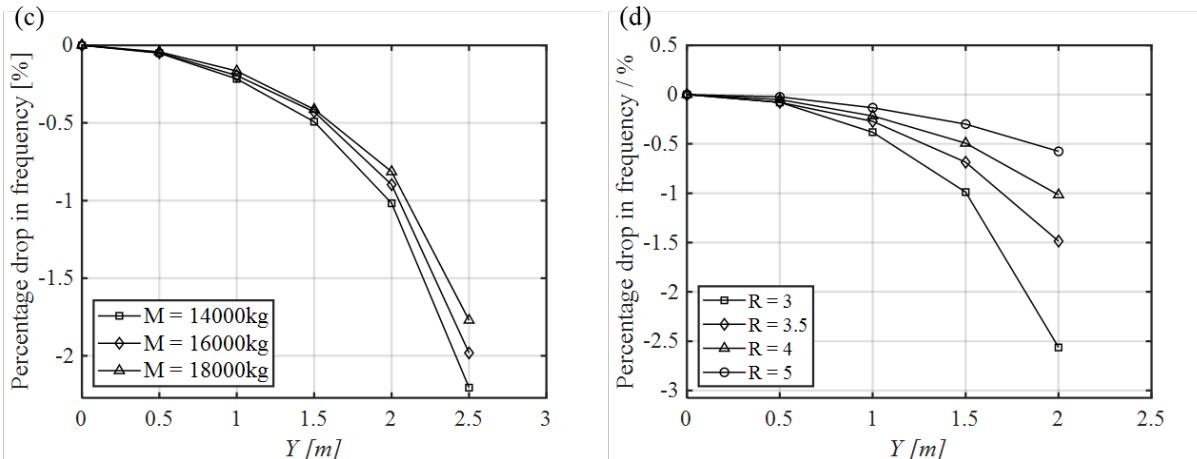

**Figure 4: The relationship between resonance frequency and amplitude of the blade at different parameters: (a)** $M = 14000kg$; $k = 210000N/m$; $L = 4m$; $R = 4m$ **(b)** $M = 14000kg$; $m = 2000kg$; $L = 4m$; $R = 4m$ **(c)** $k = 210000N/m$; $m = 2000kg$; $L = 4m$; $R = 4m$ **(d)** $M = 14000kg$; $k = 210000N/m$; $m = 2000kg$; $L = 4m$.

**2.3 Analysis of amplitude-frequency characteristics of the model**

The dynamic differential equations of the blade-virtual masses test system, established through the Lagrange method, are highly complex and can only be resolved numerically to derive the correlations among the relevant parameters and the resonance frequency of the test system. To further analyze the nonlinear amplitude-frequency characteristics of the test system, it is necessary to create a theoretical model of the test system based on nonlinear dynamics (Liu et al., 2001). According to linear vibration theory (Liu et al., 2019), the factors that primarily influence the inherent characteristics of a linear system are the inertial force term and the elastic force term. In fact, the inherent characteristics of the blade-virtual masses test system are primarily determined by the inertial force term associated with the introduction of virtual masses and the response of the blade Thus, the weakly nonlinear dynamic equation of the blade-virtual masses test system in Eqs. (13) can be approximated as:

$$(M + m)f(y)\ddot{y} + c\dot{y} + ky = F_0 \cos(\omega t + \theta) \tag{14}$$

Where: $f(y) = 1 + \varepsilon_1 y + \varepsilon_2 y^2 + \varepsilon_3 y^3 + \varepsilon_4 y^4$; $c = 2\zeta(M + m)\omega_n$; $k = (M + m)\omega_n^2$; $F_0 = Bk$; $\varepsilon_1$、 $\varepsilon_2$、 $\varepsilon_3$、 $\varepsilon_4$ - Small parameters related to $M$、 $m$、 $L$ and $R$; $\zeta$ - Damping ratio; $\omega_n$ - Natural frequency; $\omega$ - Excitation frequency; $\theta$ - Phase difference between steady-state response and excitation.

Ignoring the small parameters, Eqs. (14) is transformed into the vibration equation of a linear system. This means that the linear system is derived from the original nonlinear system. To quantitatively analyze the modal characteristics of the test system, the approximate analytical method can be employed by considering the nonlinear factor as a perturbation to the linear system, yielding an approximate analytical solution for the nonlinear system. Among various approximate analytical methods, the harmonic balance method is particularly notable due to its clear conceptual foundation. It expands both the excitation term and the solution of the equation into a Fourier series. From a physical perspective, the coefficients of the harmonic terms of the same order at both ends of the dynamic equation must be equal to maintain a balance between the excitation and inertia forces. When the condition of the test system is determined, the value of the small parameter in Eqs. (14) is also determined.

For the blade-virtual masses test system, it is assumed that its steady-state response is still periodic, but the resonance frequency is different from the natural frequency of the derived system. The basic solution is expanded into the Fourier series of the excitation frequency and the fundamental component is retained. The response of the system as Eq. (15) indicates.

$$y(t) = Y_0 \cos(\omega t) \tag{15}$$

Where: $Y_0$ - Amplitude of blade steady-state response.

By substituting Eq. (15) into Eq. (14) and applying the triangle transform and harmonic balance to eliminate the phase difference $\theta$ to achieve the relationship between the amplitude and frequency of the test system, as Eq. (16) indicates.

$$\left[1 - s^2\left(1 + \frac{3}{4}\varepsilon_2 Y_0{}^2 + \frac{10}{16}\varepsilon_4 Y_0{}^4\right)\right]^2 + (2\zeta s)^2 = \left(\frac{B}{Y_0}\right)^2 \tag{16}$$

Where: $s = \omega/\omega_n$.

227         According to Eq. (16), The amplitude-frequency and phase-frequency characteristics of the nonlinear system can be
obtained, as Eq. (17) indicates.
$$
\begin{cases}
\dfrac{Y_0}{B} = \dfrac{1}{\sqrt{\left[1-s^2\left(1+\frac{3}{4}\varepsilon_2 {Y_0}^2+\frac{10}{16}\varepsilon_4 {Y_0}^4\right)\right]^2+(2\zeta s)^2}} \\
\theta = \arctan\left[\dfrac{2\zeta s}{1-s^2\left(1+\frac{3}{4}\varepsilon_2 {Y_0}^2+\frac{10}{16}\varepsilon_4 {Y_0}^4\right)}\right]
\end{cases}
\tag{17}
$$

230         When $\varepsilon_2 = \varepsilon_4 = 0$, Eq. (17) describes the amplitude-frequency characteristics of a linear system, as shown in Fig. 5.
When the small parameters are non-zero, the amplitude-frequency characteristic curve of the nonlinear system is depicted in
Fig. 6. Similar to forced vibrations in linear systems, nonlinear systems also exhibit similar amplitude-frequency characteristic
curves. However, the backbone of the support curve clusters is not straight but inclined. This backbone curve represents the
variation of the free vibration frequency of the nonlinear system with respect to the amplitude when there is no external
excitation (Liu et al., 2001). By setting B = 0.1 and $\zeta = 0$ in Eq. (16), the equation for this backbone curve can be obtained, as
Eq. (18) indicates.
$$\omega^2 = \frac{\omega_n^2}{\left(1+\frac{3}{4}\varepsilon_2 {Y_0}^2+\frac{10}{16}\varepsilon_4 {Y_0}^4\right)} \tag{18}$$

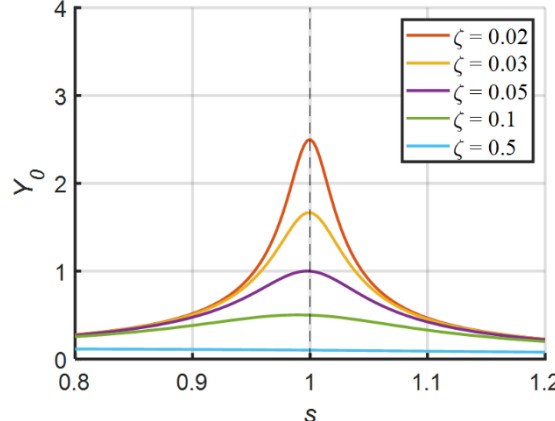

**Figure 5: Amplitude-frequency characteristic curve of a linear system**

240         Eq. (18) shows that the resonance frequency of the blade-virtual masses test system decreases with the increase of the
amplitude of the blade and there exists the nonlinear relationship between the square of the frequency ratio and the amplitude.
Figure 6 shows that the small parameters in the inertial force term will affect the frequency of free vibration. As these small
parameters decrease, the amplitude-frequency characteristic curve of a nonlinear system approaches that of a linear system,
and the backbone curve approaches a value close to 1.

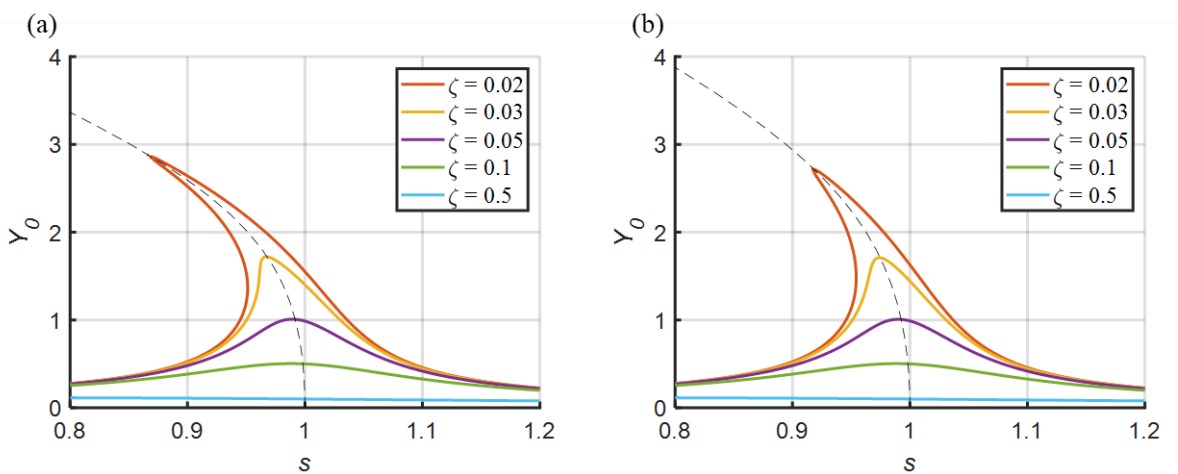


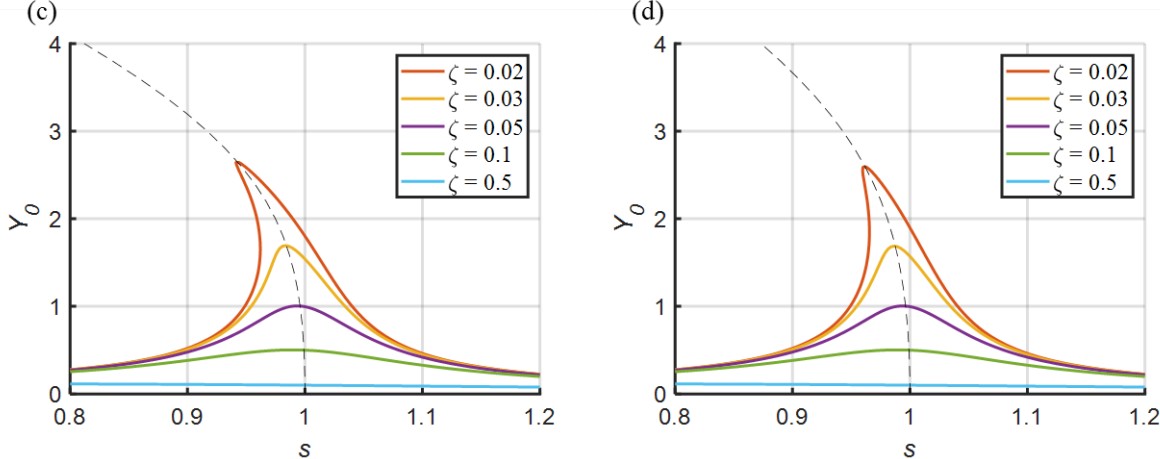

**Figure 6: Amplitude-frequency characteristic and the backbone (represented by the black dashed line) of the blade-virtual masses testing system: (a)** $B = 0.1 \setminus \frac{3}{4}\varepsilon_2 = 0.01 \setminus \frac{10}{16}\varepsilon_4 = 0.002$; **(b)** $B = 0.1 \setminus \frac{3}{4}\varepsilon_2 = 0.01 \setminus \frac{10}{16}\varepsilon_4 = 0.001$; **(c)** $B = 0.1 \setminus \frac{3}{4}\varepsilon_2 = 0.005 \setminus \frac{10}{16}\varepsilon_4 = 0.001$; **(d)** $B = 0.1 \setminus \frac{3}{4}\varepsilon_2 = 0.005 \setminus \frac{10}{16}\varepsilon_4 = 0.0005$.

     Figure 6 shows the influence of different small parameters on the amplitude-frequency characteristics of the system. In fact, specific small parameter values mean specific working conditions, that is, when the virtual mass related parameters (such as $L \setminus R \setminus m$) are determined, the amplitude-frequency characteristics of the system will also be determined. Therefore, as long as the setups are determined, the dynamic characteristics of the test system will be determined, whether it is a uniaxial axis test or a biaxial test.

     In addition, the amplitude hopping phenomenon, also known as dynamic bifurcation, also appears in Figure 6. In fact, there is no obvious dynamic bifurcation phenomenon in the fatigue test, because the nonlinearity of the system is weak, and the amplitude of the blade is limited by the size of the mechanism. Moreover, when the influence of blade amplitude on the resonance frequency of the system is discussed in the following paper, more attention is paid to the backbone curve in the shape of the black dotted line in Fig. 6.

## 3 Dynamic simulation analysis

To validate the nonlinear characteristics of the blade-virtual masses test system that has been established, it is necessary to utilize multi-body dynamics simulation software ADAMS to create a realistic blade model for analysis. Based on the sectional properties and tuning masses of the blade, ADAMS can be employed for modeling and analyzing the blade-virtual masses system. ADAMS can perform modal analysis and transient sweep frequency analysis to obtain the changing characteristic of the test system under various operating conditions. As the foundation for other dynamics analysis, modal analysis is used to determine the modal characteristics of structures. Regarding the weakness of modal analysis function in the software, which cannot consider the effects of the response on the modal characteristics of the system, it is necessary to take further transient sweep-frequency analysis to obtain the resonance characteristics of the system.

### 3.1 Simulation Modelling

To verify that the simplified equivalent theoretical model can reflect the characteristics of actual test system, the simulation model is established in software. Generally, only the cross-section stiffness (flap-wise and edge-wise) and linear density are considered in the simulation model (Post et al., 2016), because the torsional natural frequency is much higher than the natural frequency in the direction of flap-wise and edge-wise, it is difficult to stimulate large torsional deformation. The root of the blade was set as a fixed constraint to simulate the cantilever beam condition similar to when the blade is mounted on the test rig. The equivalent damping ratio of the blade changes during vibration, resulting in a change in the resonance frequency of the test system (Lee., 2018; Liu et al., 2019). In order to accurately assess the influence of virtual masses on the characteristics of the testing system, aerodynamic damping is not considered in the simulation model. The blade model was built in the

simulation software based on the parameters mentioned above, as shown in Fig. 7(a).

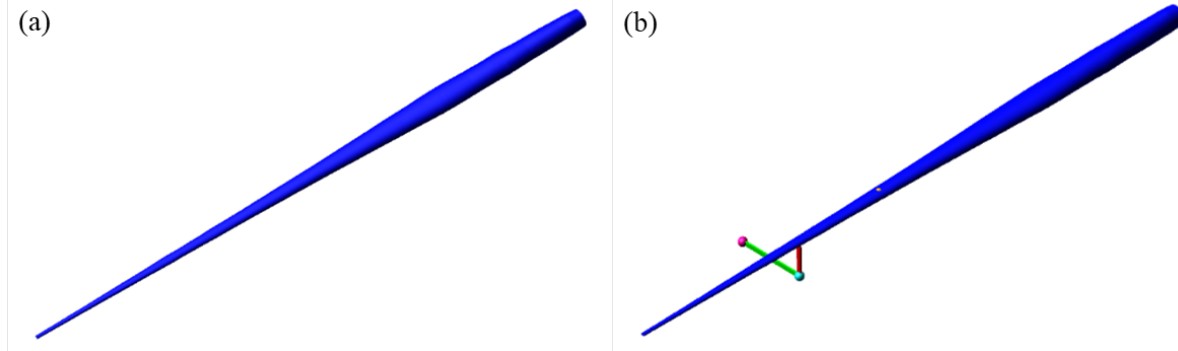

**Figure 7: Dynamics simulation model of test system: (a) The blade simulation model (b) The blade-virtual masses simulation**
**model(flap-wise)**
**3.2 Model validity verification**
To ensure the applicability and rationality of the model, modal analysis is carried out and compared with the transfer-
matrix method (TMM) and the test data, as shown in Table 1. The transfer matrix method is an approximate theoretical method
used to calculate the natural frequencies and modes of systems with chain structures. The transfer matrix method separates the
structure with inertia and elasticity and obtains the relationship between the discrete elements. The natural frequencies and
modes of the systems can be solved according to the boundary conditions. The transfer matrix method belongs to the physical
discrete method of continuous system, which is suitable for numerical solution of blade model. The blades in Table 1 were all
subjected to actual modal tests, and the obtained frequency data are obtained from the frequency domain analysis of actual test
data. The actual blade modal test was carried out by hammer method. It can be seen that the simulation model of the test system
has good applicability, with an error in the modal frequency of less than 4%.
**Table 1. Comparison of natural frequencies calculated by various methods**

| Flap-wise | 84m | | 94m | |
|---|---|---|---|---|
| Method | 1st modal frequency [Hz] | Error [%] | 1st modal frequency [Hz] | Error [%] |
| Test | 0.394 | - | 0.365 | - |
| TMM | 0.397 | +0.7 | 0.349 | -4.38 |
| Simulation | 0.404 | +2.54 | 0.377 | +3.29 |
| Edge-wise | 84m | | 94m | |
| Method | 1st modal frequency [Hz] | Error [%] | 1st modal frequency [Hz] | Error [%] |
| Test | 0.590 | - | 0.571 | - |
| TMM | 0.604 | +2.37 | 0.561 | -1.75 |
| Simulation | 0.610 | +3.34 | 0.589 | +3.15 |

**3.3 Simulation setup**
With the purpose of demonstrating the nonlinear effects of rotating virtual masses on the testing system, it is necessary to add
virtual masses based on the blade model, as shown in Fig.7 (b). The values of the tuning masses are shown in Table 2 and the
section properties of the blades are shown in Fig. 8. The position and values of the tuning masses are provided by the blade
manufacturer. Virtual masses elements and exciting force are added at 62% and 49% of the 84m blade length in the flap-wise
and edge-wise directions respectively. Similarly, virtual masses elements and exciting force are added at 63% and 52% of the
94m blade length in the flap-wise and edge-wise directions respectively (masses marked in black italics in Table 2). The
constraints for the seesaw, push rod, and virtual masses are set according to Fig. 1, where the rotation center of the seesaw is
set as the revolute pair and the seesaw and push rod are set as the rigid light rod. To evaluate and verify the effects of virtual
masses installation parameters and blade response on the vibration characteristics of the test system, not only the effects of
radius of the seesaw and blade response on the resonance frequency, but also the effects of radius of the seesaw on the load
distribution of the blade with similar amplitude are analyzed through simulation.
**Table2. Blade additional masses of 84m and 94m blade**

| 84m | | | 94m | | |
|---|---|---|---|---|---|
| Location | Flap-wise masses [kg] | Edge-wise masses [kg] | Location | Flap-wise masses [kg] | Edge-wise masses [kg] |
| 26% | | 2835 | 42% | 3000 | 3000 |
| 36% | | 3147 | 52% | | *4075* |
| 49% | 6120 | *4075* | 63% | *1116* | |
| 62% | *1117* | | | | |

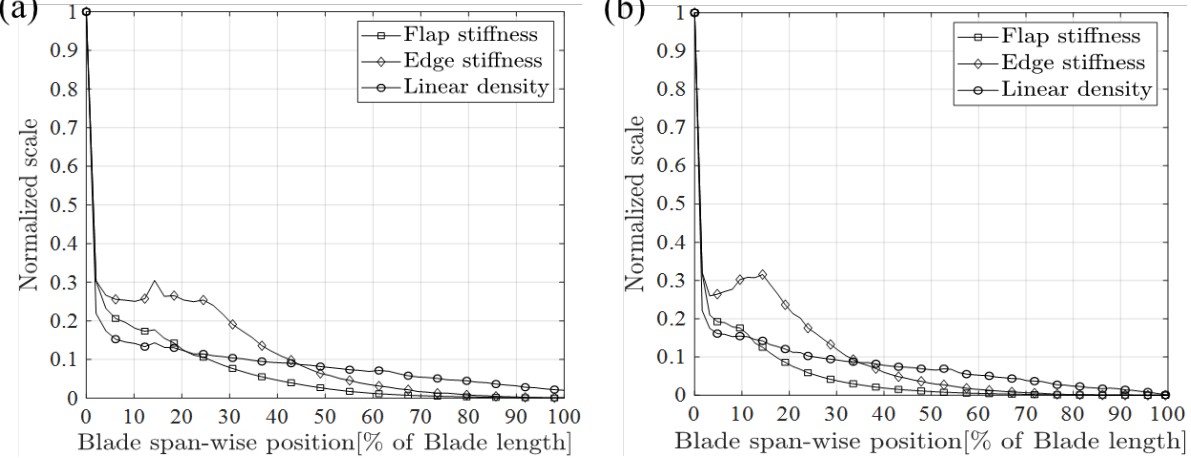


**Figure 8: Section properties of the blade: (a) 84m blade (b) 94m blade**

## 4 Results


According to the backbone in the amplitude-frequency characteristic curve of the blade-virtual masses test system, when the
operation condition determined, the square of the resonance frequency and the blade amplitude satisfy the relationship in Eqs.
(18). Thus, correlated simulation results are fitted using relevant functions to verify the relationship.

### 4.1 Effects of virtual masses on uniaxial test

### 4.1.1 Effects of blade amplitude on resonance frequency

Set $R$ = 4m and $L$ = 4m and investigate the variation of the resonance frequency of test system under different amplitudes.
Sweep-frequency analysis is performed on the 84m and 94m blades in flap-wise and edge-wise directions respectively to obtain
the resonance frequencies of the test system under different steady-state amplitudes while the results are fitted according to
Eqs. (18), as shown in Fig. 9. In addition, the degree of fit is expressed by goodness of fit $R^2$. The sweep frequency range is
defined as a bandwidth of 0.02Hz near the first natural frequency in the flap-wise or edge-wise direction, with an action
time of 1E4s and a resolution of 2e-6 Hz/sec. The frequency spectrum of the displacement of the exciting point of the
blade under the sweeping excitation is analyzed, and the frequency corresponding to the peak point is the resonance
frequency. The mechanism might reach the geometric limit of the push-rod parallel to the seesaw, so the limit
requirements of the mechanism need to be considered.
When amplitude of the blade is small, the percentage drop in resonance frequency is small. When amplitude of the blade
is large, the resonance frequency decreases nonlinear faster. When the blade amplitude in flap-wise direction reaches 2.6m,
the resonance frequency of the 84m and 94m blades decreases by approximately 2.0%; When the blade amplitude in edge-
wise direction reaches 2.2m, the resonance frequency of the 84m and 94m blades decreases by only approximately 1.1%. Due
to the limitation of resonance frequency extraction precision in sweep frequency analysis, the fitting degree of data is
affected. However, it is still acceptable at the large amplitude of the blade. Combined with the actual test requirements,
we should pay more attention to the conditions of large amplitude.

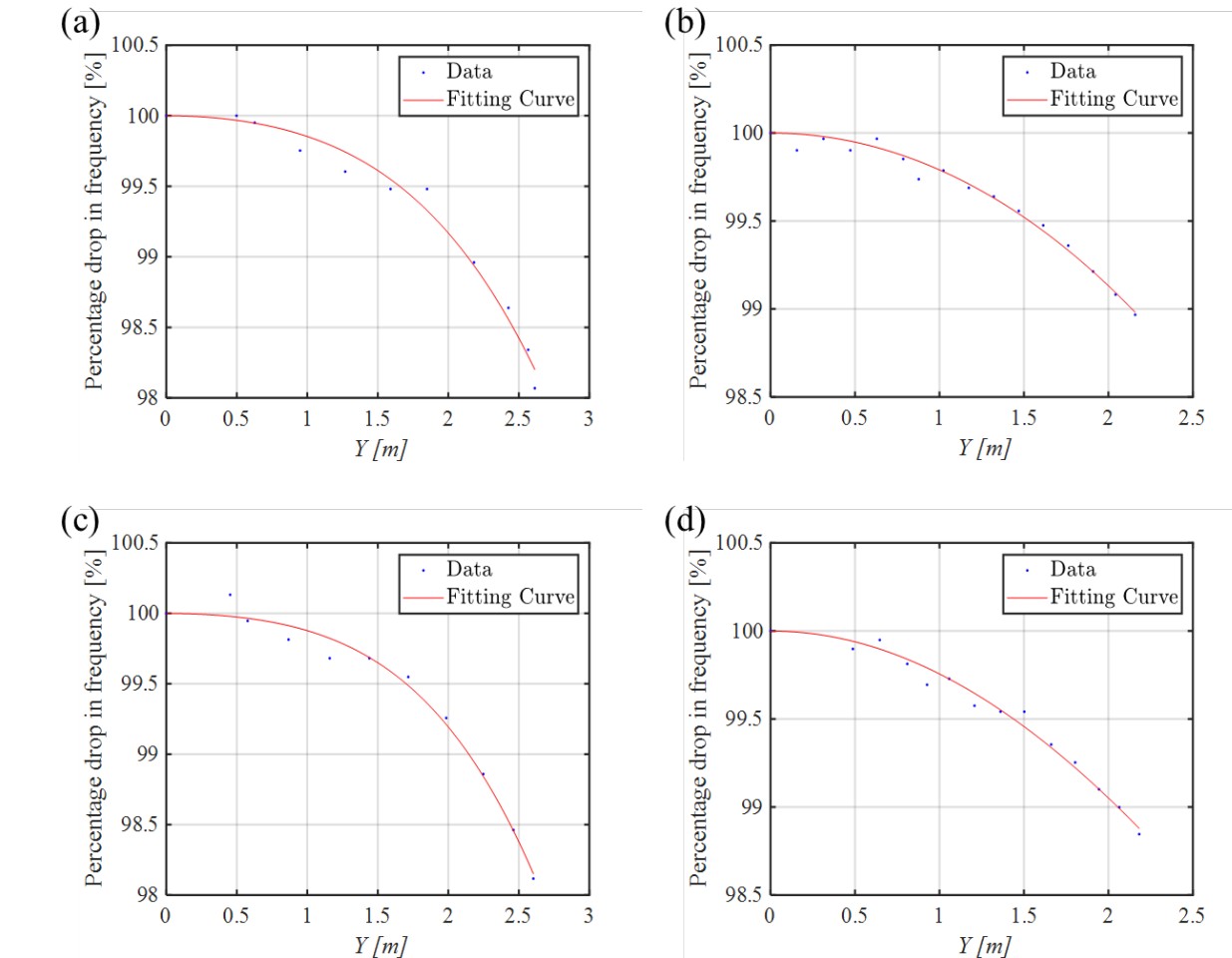



**Figure 9: Relationship between amplitude and percentage drop in resonance frequency: (a) 84m blade in flap-wise direction ( $R^2 =$**
**$0.9814$); (b) 84m blade in edge-wise direction ( $R^2 = 0.9829$); (c) 94m blade in flap-wise direction ( $R^2 = 0.9861$); (d) 94m blade**
**in edge-wise direction ( $R^2 = 0.9831$)**

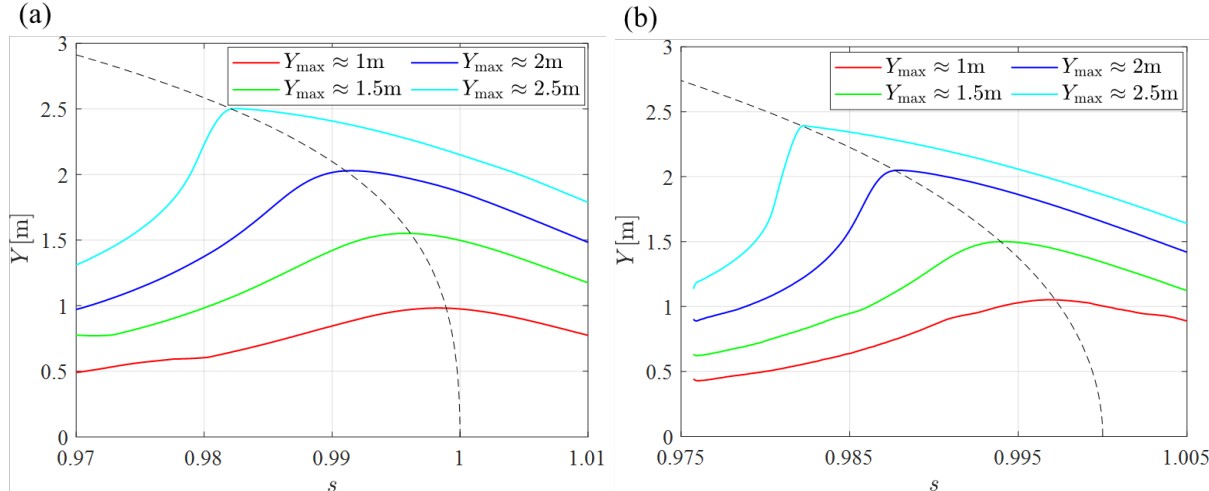


**Figure 10: The sweep spectrum of the blade under different target amplitudes**
Taking 94m blade as an example, the sweep spectrum of the blade under different target amplitudes is shown in Fig. 10.
It can be seen from Fig.10 that different excitation frequencies cause different blade responses. The resonance frequency of
the system decreases with the increase of the maximum amplitude of the blade. This also verifies the applicability of the
approximate amplitude-frequency properties obtained by the theory (Fig. 6 has a backbone curve similar to Fig. 10).
**4.1.2 Effects of radius of the seesaw on resonance frequency and load distribution**
Considering the actual test setup, the blade amplitude in flap-wise direction is set to be about $Y$=2m and the length of the push
rod is $L$=4m; the blade amplitude in edge-wise direction is about $Y$=1m and the length of the push rod is $L$=4m. The sweep-
frequency analysis of the 84m and 94m blades in flap-wise and edge-wise directions is carried out respectively to obtain the
resonance frequency of the test system. According to Eqs. (18), appropriate function (Eqs. (19)) is selected to fit the results, as
shown in Fig. 11. Eqs. (19) is a function selected according to the degree of best fit. Considering equations (18) and (19), the
small parameters encompass the influence of radius of the seesaw, which can be approximated by an exponential function. A
larger radius of the seesaw results in a smaller decrease in the resonance frequency. Conversely, when the rotation radius of
the seesaw is small, the resonance frequency experiences a nonlinear decrease. With R = 3m, the drop in the resonance
frequency of the 84m and 94m blades is approximately 1.6% in the flap-wise direction. Likewise, with R = 2m, the drop in the
resonance frequency is only approximately 1.1% in the edge-wise direction.
$$\omega^2 = \frac{\omega_n^2}{(1+ae^{-bR})} \qquad (19)$$
Where: $a$、$b$ - parameters in exponential function.

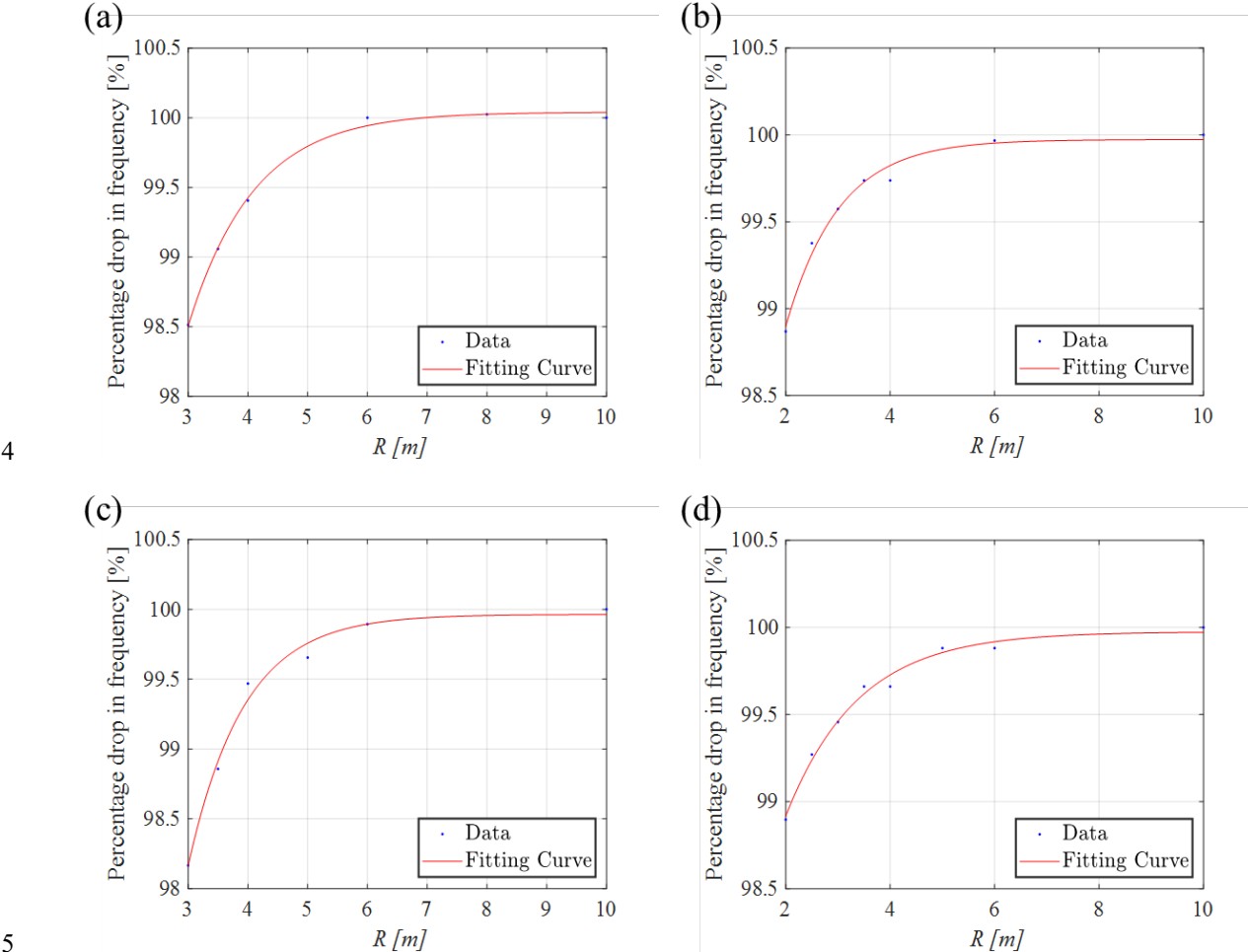

**Figure 11: Relationship between radius of the seesaw and percentage drop in resonance frequency: (a) 84m blade in flap-wise**
**direction ( $R^2 = 0.9973$); (b) 84m blade in edge-wise direction ( $R^2 = 0.9786$); (c) 94m blade in flap-wise direction    ( $R^2 =$**
**0.9884); (d) 94m blade in edge-wise direction ( $R^2 = 0.9890$)**

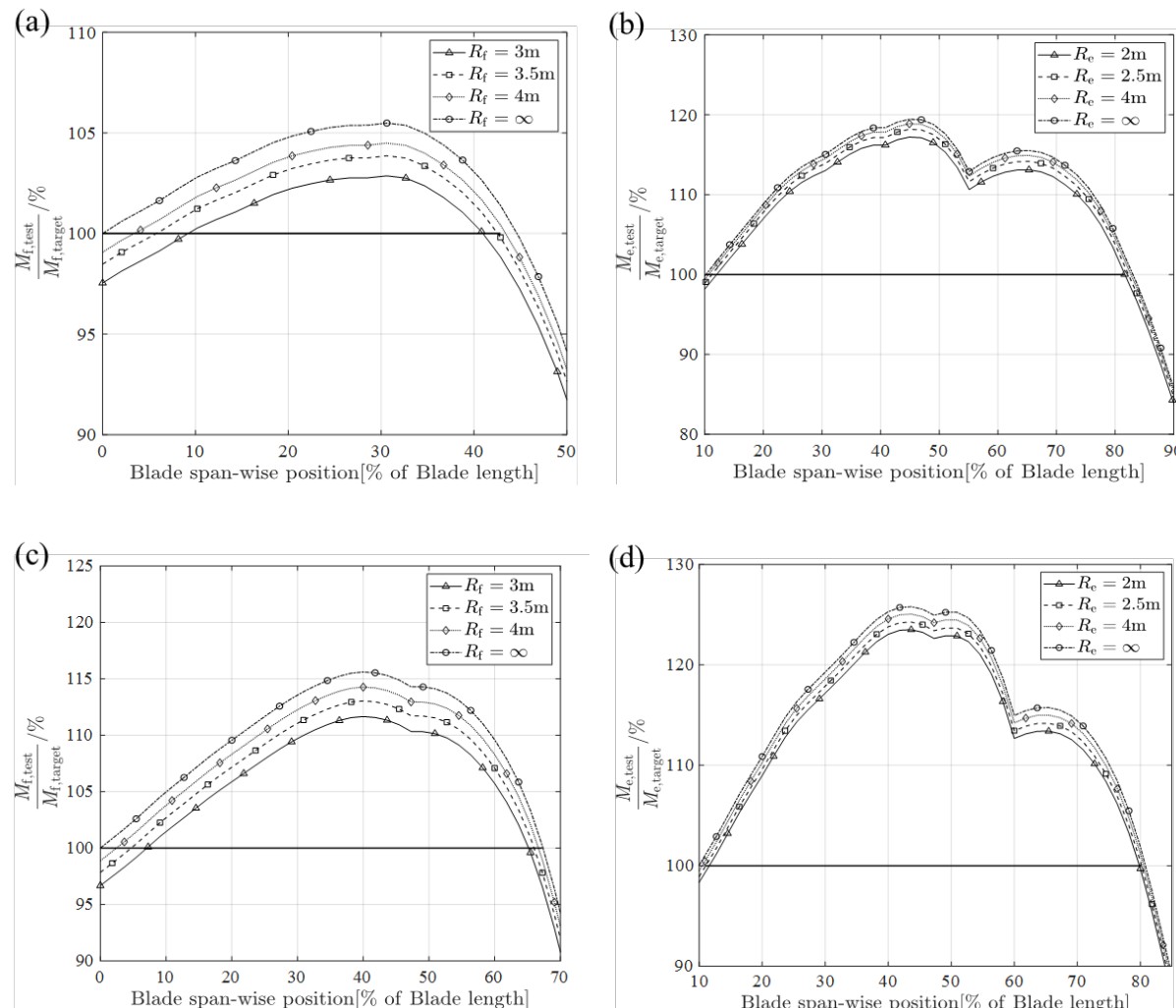

**Figure 12: Relationship between radius of the seesaw and blade load distribution: (a) 84m blade in flap-wise direction (b) 84m blade in edge-wise direction (c) 94m blade in flap-wise direction (d) 94m blade in edge-wise direction**

In order to compare the influence of nonlinearity on the blade load distribution, the blade bending moment distribution can be calculated by using constant displacement of the exciting point and inertial load provided by virtual mass motion. The excitation position is the same as installation position of the virtual masses closest to the tip of the blade, and the specific values are shown in Table 2. The excitation frequency is the resonance frequency of the respective vibration direction, which is obtained by the sweep frequency analysis.

The radius of the seesaw influences the characteristics of the testing system and alters the distribution of blade loads, as shown in Fig. 12. In the case of $R = \infty$, the virtual masses shift from rotation to translation in the uniaxial test, effectively simulating additional masses that are directly fixed onto the blade. Consequently, there is an approximate 3% decrease in the overall load distribution in the flap-wise direction, the area which is actually fully tested will be reduced. Given the roughly similar amplitudes, lower resonance frequency results in reduced inertial loads on the blade. Therefore, compensatory measures such as increasing the excitation level are necessary during the actual test. However, this requires more powerful excitation equipment.

**4.2 Effects of virtual masses on biaxial test**

In Section 4.1, only the effect of virtual masses on the uniaxial test is considered, which can intuitively see the influence of independent parameters on the vibration characteristics of the test system and blade load distribution from the uniaxial model. However, it is not enough to consider only the uniaxial vibration, but also the effect of virtual masses on the system in the biaxial vibration. In the biaxial test, the blade has a complex spatial trajectory, and the test system will be affected by multiple

nonlinear parameters at the same time. To find the resonance frequency of the two directions, it is necessary to use the
simulation software for iterative calculation.
Taking 94m blade as an example, virtual masses are applied in both flap-wise and edge-wise directions. Modal analysis
and frequency sweep analysis are used to obtain the frequencies at which specific excitations are applied to the test system.
Combined with the actual working conditions, the flap amplitude at 63% position of the blade is about 2m, and the edge
amplitude at 52% position of the blade is about 1m. The resonance frequencies under different conditions are shown in Table
3, with R = 4m and L = 4m. In fact, the oscillations in flap-wise and edge-wise direction must not be evaluated separately
as they influence each other, so the resonance frequency of the blade in each direction is obtained by sweeping frequency
iteration. Fig. 12 shows the spatial trajectory of the blade under the action of different virtual mass mechanisms. The
results show three main characteristics: 1) Under the same exciting force, the resonance frequency of the two directions
in the biaxial test is lower than that of the uniaxial test, which indicates that the virtual masses affect both vibration
directions. 2) Compared with the ideal working condition, the virtual masses will deform the space trajectory of the blade
(even considering the structural torsion of the blade), which is determined by the motion characteristics of mechanism.
In addition, the deformation of the trajectory may bring higher requirements for the actual damage assessment of the
blade. 3) Under the same exciting force, the difference between the average flap amplitude of the blade using the rotating
virtual mass mechanism and the average flap amplitude under the ideal condition is 9%, and the difference between the
average edge amplitude and the average edge amplitude under the ideal condition is nearly 11%, as shown in Table 4.
Combined with the effect of reduced resonance frequency and amplitude, the biaxial load distribution level of the blade
will be further reduced compared with the uniaxial test, which means that more energy input is required.

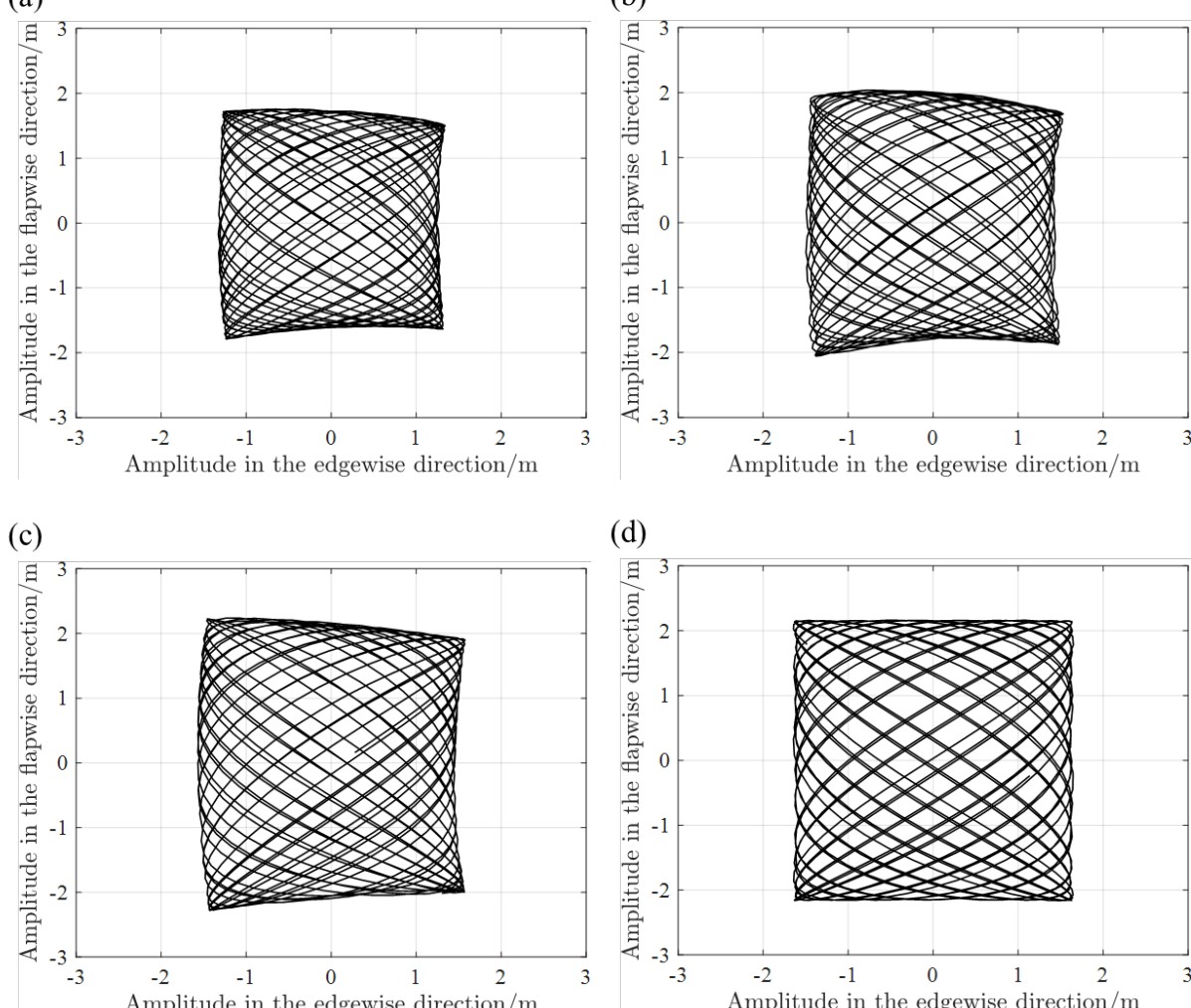



**Figure 13: Biaxial trajectory of blade-virtual masses test system with same exciting force (at 63% of the blade position): (a)**
**Natural frequency excitation; (b) Resonance frequency excitation(Rotation);(c)Resonance frequency excitation(Actual**
Translation);(d)Resonance frequency excitation(Ideal Translation)
**Table3. Biaxial excitation parameters of 94m**

| Virtual masses and exciting point | | Natural frequency [Hz] | Uniaxial resonance frequency (Rotation) [Hz] | Sweep-frequency analysis | | |
|---|---|---|---|---|---|---|
| | | | | Biaxial resonance frequency [Hz] | | |
| Position [%] | Force [N] | | | Rotation | Actual translation | Ideal translation |
| Flap 63% | 3800 | 0.377 | 0.373 | 0.369 | 0.375 | 0.377 |
| Edge 52% | 7000 | 0.589 | 0.587 | 0.583 | 0.587 | 0.589 |

**Table4. Biaxial amplitude of 94m**

| | Biaxial average amplitude [m] | | |
|---|---|---|---|
| | Rotation | Actual translation | Ideal translation |
| Flap | 1.923 | 2.113 | 2.154 |
| Edge | 1.447 | 1.511 | 1.630 |

## 5 Conclusion
The nonlinear effect of virtual mass device on blade test system is discussed in this paper. In actual working conditions, the
test system is limited by the size of the virtual mass mechanism and the amplitude of the blade, and its resonance characteristics
will be changed. This paper firstly analyzed the nonlinearity of the system resonance characteristics from the mechanism of
the change of the inertia force of the virtual mass, and established a blade uniaxial theoretical model to explore the influence
of the amplitude of the blade and the size of the seesaw on the resonance frequency. Based on the above content, the
approximate nonlinear amplitude-frequency characteristic curve of the test system is obtained. Then the software is used to
simulate the two blades by the transient sweep method, and the applicability of the theoretical model is verified.
For the uniaxial theoretical model, the increase of blade amplitude、the shortening of seesaw size and the increase of
counterweight mass will reduce the resonance frequency in the main vibration direction. However, the uniaxial simulation
results of two blades show that the amplitude of the blade or the size of the seesaw have limited influence on the resonance
frequency. For example, when the size of the mechanism is unchanged ($L = R = 4m$), only the influence of blade amplitude
on the system is considered. When the amplitude of the flapping direction increases to 2.6m, the resonance frequency in this
direction decreases by nearly 2% compared with the natural frequency; Combined with the actual working conditions, when
the amplitude of the flap-wise direction is maintained about 2m, the length of the seesaw is shortened to 3m, and the resonance
frequency is reduced by nearly 2%. The target amplitude of the edge-wise is usually small compared with the flap-wise
direction, so shortening the length of the seesaw only reduces the resonance frequency by 1.1%. In the case of the same
amplitude, the shortening of seesaw length will reduce the blade load distribution level, and the flap-wise load level will
decrease by up to 3% at most. Due to the small amplitude of the edge-wise, the load level in this direction does not drop
significantly.
Although the nonlinear factors have less influence on the uniaxial test, they have more influence on the biaxial test. Under
the same excitation force and approximate target amplitude (Y≈2m), the rotating virtual masses induces lower resonance
frequency in biaxial vibration than in uniaxial test (Flap-wise direction: 1.06% decrease in uniaxial vibration, 2.12% decrease
in biaxial vibration; Edge-wise direction: 0.34% decrease in uniaxial, 1.02% decrease in biaxial). In addition, under the same
exciting force, the difference between the average flap amplitude of the blade using the rotating virtual mass mechanism
and the average flap amplitude under the ideal condition is 9%, and the difference between the average edge amplitude
and the average edge amplitude under the ideal condition is nearly 11%. Furthermore, the virtual masses mechanism can

also cause the deformation of the space trajectory envelope of the blade. Under the combined action of many factors, the nonlinear effect will be further strengthened.

In conclusion, the virtual masses mechanism will bring nonlinear effect to the test system due to its own motion characteristics, and the nonlinear factors mainly include the amplitude of the blade, the size of the mechanism and the mass of the counterweight. In the case of small amplitude, the nonlinear effect is not obvious and has not great influence on the blade load level. In the biaxial large amplitude test, the nonlinear effect is enhanced and the blade trajectory is deformed. The resonance frequency of the system will be further reduced. Under the same excitation, the actual blade amplitude is less than the target amplitude. These characteristics mean that biaxial test requires larger excitation equipment and higher requirements for blade damage calculation and load formulation.

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

**Code and data availability.**
The data that support the findings of this research are not publicly available due to confidentiality constraints.
**Author contributions.**
JS conceptualized and defined the requirements for the method developed. AZ supervised the work. JS and TD developed
the model code and performed the simulations. JS prepared the manuscript with contributions from all co-authors.
**Competing interests.**
The authors declare that they have no conflict of interest.
**Financial support.**
Blade data of this research has been supported by Aeolon Technology Co., L