# Peer review of "Nonlinear Vibration Characteristics of Virtual Mass Systems for Wind Turbine Blade Fatigue Test"

_Wind Energy Science, 2023_

## Author Comment (AC2)

Dear Reviewer #1:

Thank you for your comments and suggestions. Those comments are all valuable and very helpful for revising and improving our paper, as well as the important guiding significance to our researchers. We have studied comments carefully and have made correction which we hope meet with approval. Now I response the comments with a point by point. Full details of the files are listed. We sincerely hope that you find our response and modifications satisfactory and that the manuscript is now acceptable for publication.

Responds to the reviewer's comments:

**Comment 1:** Grammar errors should be noted (e.g. in abstract Lines 10-11: the nonlinear amplitude-frequency characteristics of the test system IS analyzed...). Please check.

**Response 1:**

We agree with the comment and thanks for pointing out our grammar errors. In addition to fixing the errors in comment 1, we have also checked the rest of the text. The contents of the modification are as follows:

(1) Lines 10-11: the nonlinear amplitude-frequency characteristics of the test system are analyzed theoretically based on the nonlinear vibration theory.

(2) Lines 49,52,205,285: Revise "effect" to "effects".

(3) Lines 60-61: In the biaxial fatigue test, the additional masses decouple the biaxial load by seesaw, and the additional masses are called virtual masses, as shown in Fig. 1 (b). In this installation condition, the inertia force generated by the virtual masses only acts in the direction of an individual blade mode.

(4) Lines 2,30,62-67,72,74,78,128,137-138,184,207-208: Revise "mass" to "masses".

(5) Lines 265: Given the roughly similar amplitudes, lower resonance frequency results in reduced inertial loads on the blade.

**Comment 2:** In Lines 43-46, authors state: "Therefore, IWES conducted further research, designed a device to convert virtual masses from translation to rotation……" Where is the corresponding reference of this research? Please check it.

**Response 2:**

Thank you for pointing this out, we actually put the reference in the next sentence. However, to ensure the rigor of the article, we have adopted your suggestion to add the corresponding reference of research of IWES. The modification content is as follows:

Lines 43-46: "Therefore, IWES conducted further research, designed a device to convert virtual masses from translation to rotation, and applied it to the biaxial fatigue test which has a frequency ratio of 1:1 (Melcher et al., 2020)"

Corresponding reference:

*[12] Melcher, D., Petersen, E., and Neßlinger, S.: Off-axis loading in rotor blade fatigue tests with elliptical biaxial resonant excitation, J. Phys. Conf. Ser, 1618(5): 052010, https://doi.org/10.1088/1742-6596/1618/5/052010, 2020.*

**Comment 3:** In Lines 46-47, authors state: "the inertia force generated by rotating virtual masses is different from that generated by translational virtual masses." Please explicitly illustrate the difference between these two setups and explain its effects on inertia force. What is the motivation of studying nonlinear vibration characteristics of wind turbine blades based on Virtual mass match.

**Response 3:**

Thank you very much for your suggestions. We will add the explanation of text and theoretical analysis to the article. The modification content is as follows:

(1) The difference between these two setups

Add content to the original lines 46-47: "In fact, in the view of the motion characteristics, the inertia force generated by rotating virtual masses is different from that generated by translational virtual masses. The translational virtual masses move synchronously with the blade, which behave like a mass acting in just one direction from a numerical standpoint. The translational virtual masses have the same motion characteristics as the additional tuning masses. Therefore, although the virtual mass is not on the blade, the inertia force generated by it and the inertia force generated by the additional tuning masses are in the same direction and magnitude. The rotating virtual masses are limited by the constraints of the seesaw, and its motion path is the rotating motion around the center of the seesaw. Therefore, the direction and magnitude of the inertial force generated by the rotation of the virtual mass will change, and it is not equivalent to the translational virtual masses."

If the text is not clear, it can be understood with the picture below.

[Figure]

Corresponding reference:

*[10] Post, N. and Bürkner, F.: Fatigue Test Design: Scenarios for Biaxial Fatigue Testing of a 60-Meter Wind Turbine Blade, Tech. rep., National Renewable Energy Laboratory, Golden, CO, USA, https://doi.org/10.2172/1271941, 2016.*

*[17] Falko, B.: Biaxial Dynamic Fatigue Tests of Wind Turbine Blades, Ph.D. thesis, Leibniz University Hannover, Germany, https://publica.fraunhofer.de/handle/publica/283519, 2020.*

(2) The explanation of the effects on inertia force

In fact, the above text (1) roughly explains that the inertia forces produced by the two setups are different because of the different motions of the virtual masses. In order to more clearly explain the effects on inertial forces between the two setups, we will add the relevant content in the section 2.1 as follows:

[Figure]

**Figure2: Virtual masses setup for blade fatigue test.**

According to Fig. 2, the inertial force generated by rotating virtual masses of the blade at the maximum amplitude $Y$ can be further analyzed. The relationship of the motion between virtual mass and blade can be obtained:

$$\begin{cases} \boldsymbol{v}_m = \boldsymbol{v}_M + \boldsymbol{v}_{mM} \\ \boldsymbol{a}_m = \boldsymbol{a}_m^n + \boldsymbol{a}_m^\tau = \boldsymbol{a}_M + \boldsymbol{a}_{mM}^n + \boldsymbol{a}_{mM}^\tau \end{cases} \tag{n}$$

Where: $\boldsymbol{v}_m$ - velocity of virtual masses; $\boldsymbol{v}_M$ - velocity of blade equivalent mass; $\boldsymbol{v}_{mM}$ - relative velocity; $\boldsymbol{a}_{mM}^n$ - relative normal acceleration; $\boldsymbol{a}_m$ - the acceleration of the virtual masses; $\boldsymbol{a}_{mM}^\tau$ - relative tangential acceleration; $\boldsymbol{a}_m^n$ - normal acceleration; $\boldsymbol{a}_m^\tau$ - tangential acceleration. The blade at the maximum amplitude satisfies: $\boldsymbol{v}_M = 0$; $\boldsymbol{v}_{mM} = 0$; $\boldsymbol{a}_{mM}^n = 0$; $\boldsymbol{a}_m^n = 0$.

Depending on the direction of acceleration, Eqs. (n) can be simplified as:

$$\boldsymbol{a}_m^\tau = \boldsymbol{a}_M + \boldsymbol{a}_{mM}^\tau \tag{n+1}$$

The angular acceleration of the virtual mass at the maximum amplitude of the blade can be obtained:

$$|\alpha_m| = \frac{\omega^2 Y \cos(\beta_0)}{R \cos(\theta_0 - \beta_0)} \tag{n+2}$$

Where: $\theta$ - Rotation angle of the seesaw at the maximum amplitude of the blade; $\beta_0$ - Angle between the push rod and the main vibration direction at the maximum amplitude of the blade; $\alpha_m$ - Angular acceleration of the virtual mass at the maximum amplitude of the blade.

According to Eqs. (n+1) and Eqs. (n+2), the rotating inertia force $F_R$ generated by the rotating virtual mass at the maximum amplitude of the blade can be obtained:

$$F_R = \frac{mR^2 \alpha_m}{R} = \frac{m\omega^2 Y \cos(\beta_0)}{\cos(\theta_0 - \beta_0)} \tag{n+3}$$

The inertia force $F_{rot}$ transmitted to the main vibration direction of the blade through the push rod can be obtained:

$$F_{rot} = \frac{F_R \cos(\beta_0)}{\cos(\theta_0 - \beta_0)} = \frac{m\omega^2 Y \cos^2(\beta_0)}{\cos^2(\theta_0 - \beta_0)} \tag{n+4}$$

By comparison with Eqs. (8), it can be seen that the inertial force terms of two equations are same at the maximum amplitude of the blade. As mentioned above, the translational virtual masses are consistent with the motion state of the blade, so the inertial force generated by the translational virtual masses can be obtained based on Eqs. (n+3):

$$F_{tra} = m\omega^2 Y \tag{n+5}$$

According to Eqs. (n+4) and Eqs. (n+5), there are differences in the inertial forces acting on the blades by the two setups, which are mainly caused by the difference in the movement trajectory of masses.

$$\left\{ M + \boxed{m \frac{\cos^2 \beta}{\cos^2 (\theta - \beta)}} \right\} \ddot{y} + c\dot{y} + ky + \frac{m\dot{y}^2 \cos \beta}{\cos^4 (\theta - \beta)} \left[ \frac{\cos^2 \beta \sin(\theta - \beta)}{R} - \frac{\sin^2 \theta}{L} \right] = F(t) \tag{8}$$

(3) The motivation of studying nonlinear characteristics of blades based on Virtual masses match

The motivation is to adopt a reasonable control strategy for the nonlinearity brought by virtual masses to achieve the target damage in future research. In the blade-virtual masses system, the excitation equipment needs to have the function of automatic adjustment of resonance frequency to minimize the energy input, and the biaxial load (trajectory) envelope of the blade will change due to the change of resonance frequency and the influence of the virtual mass mechanism, so the blade's test specification may need to be adjusted to achieve the target damage.

We will add this motivation in the revised paper in line 47-48.

**Comment 4:** In Fig. 1(b), the setup of virtual masses in is different from those reported in previous works (White et al., 2004; Greaves et al., 2012; Snowberg et al., 2014; Hughes et al). It is noted that this setup introduces nonlinear terms to the test system resulting in a more complex scenario. Please explain the mechanism of the device and illustrate advantages of this device comparing with previous setups.

**Response 4:**

In the previous resonance biaxial test, a reasonable load distribution (in both directions) will be obtained by optimizing the position and mass of the counterweight installed on the blade. However, the tuning masses installed on the blade will affect the vibration characteristics (mode shape and frequency) in both flap-wise and edge-wise directions, which brings difficulty to the biaxial load match optimization, and there may be excessive overload in a certain area of the blade when choosing a compromise.

To simplify load match, the extra mechanism makes the tuning masses only act in one vibration direction (called virtual masses), and the biaxial load match is equivalent to the combination of the load match of two single axis test, as shown in Fig.1(b).

(1) The mechanism of the device

The mechanism for mounting the virtual mass consists of a push rod and a seesaw. The push rod, blade fixture, and seesaw are connected through a universal joint, and the seesaw can rotate around the center position. Masses are located at both ends of the seesaw to offset each other's gravity. After the exciting force is applied to the blade, the tuning masses move with the blade and rotate around the center of the seesaw to provide the inertia force for the blade.

[Figure]

**Figure 1: Masses match of blade fatigue test: (a) traditional tuning masses setup (b) virtual masses setup.**

(2) Advantages of this device
 ① The virtual mass only acts in one direction, which is conducive to the decoupling of biaxial loads.
 ② This device is easier to be used in real test. In the figure of **Response 3 (1)**, it's hard to achieve translational virtual masses. Because a larger and stronger platform is needed to keep virtual mass translate, which is difficult to achieve in a limited test space. In the biaxial test, the platform may interfere with the push rod, especially when the blade has a large amplitude in the edge-wise direction.

We will emphasize mechanism of the device and advantages of this device comparing with previous setups in the revised paper.

**Comment 5:** In Lines 75-79, authors state: "the inertial force of the virtual masses also affects the flap-wise direction of the blade……since the frequency of the inertial force is close to the first order modal frequency in edge-wise direction, the perturbation to the flap-wise direction is relatively small……". Is there any evidence (reference or theoretical analysis) supporting that the perturbation to the flap-wise direction is relatively small?

**Response 5:**

(1) Theoretical analysis

According to Eqs. (n+1) and Eqs. (n+2) in **Response 3** and Eqs. (8), the inertia force $F'_{rot}$ transmitted to the secondary vibration direction of the blade through the push rod can be obtained:

$$\begin{cases} F'_{rot} = \dfrac{F_R \sin(\beta_0)}{\cos(\theta_0 - \beta_0)} = \dfrac{m\omega^2 Y \cos(\beta_0)\sin(\beta_0)}{\cos^2(\theta_0 - \beta_0)} \quad (at\ the\ maximum\ amplitude\ of\ the\ blade) \\ F'_{rot}(t) = \dfrac{m\cos(\beta)\sin(\beta)}{\cos^2(\theta - \beta)}\ddot{y} \end{cases}$$

Taking 84m and 94m blade as an example, $R = L = 4m$, the blade amplitude in edge-wise direction is about $Y = 1m$, the selected parameters as shown in section 4.2. The proportion of perturbation in the flap-wise direction is:

$$\left(\frac{F'_{rot}}{F_{rot}}\right)_{max} = \frac{\sin(\beta_0)}{\cos(\beta_0)} = 0.032$$

Where: $\beta_0$ can be solved by Eqs. (2).

In addition, lines 76-77 state "since the frequency of the inertial force is close to the first order modal frequency in edge-wise direction". According to the amplitude-frequency characteristic curve, $\left(\frac{F'_{rot}}{F_{rot}}\right)_{max}$ will decrease further.

(2) Simulated analysis

Taking 84m and 94m blade as an example, $R = L = 4m$, the blade amplitude in edge-wise direction is about $Y = 1m$, the selected parameters as shown in section 4.2. From the following simulation results, the perturbation to the flap-wise direction is relatively small (0.5% of the edge-wise amplitude).

[Figure]

Figure: 84m blade amplitude of edge-wise direction

[Figure]

Figure: 84m blade perturbation amplitude of flap-wise direction

[Figure]

Figure: 94m blade amplitude of edge-wise direction

[Figure]

Figure: 94m blade perturbation amplitude of flap-wise direction

**Comment 6:** In section 2.1, the equivalent dynamic model of the blade-virtual masses test system is established with only edge-wise degree of freedom considered. Considering that this kind of device is designed for biaxial fatigue test, why is the flap-wise degree of freedom not included?

**Response 6:**

Your question is very reasonable. In our previous study, we also considered this question, but we did not consider another directional degree of freedom for the following reasons:

(1) In the uniaxial model, the perturbation in the other direction is relatively small, as shown in Figure 2 and **response 5**.

(2) We expect to use a uniaxial model to analyze the blade-virtual masses system, so that readers can understand the nonlinear effects introduced by the virtual masses more easily. The reason for this is that the biaxial test will make blade move more complex, and the nonlinear vibration characteristics of any vibration direction will be affected by many factors at the same time, so it is difficult to analyze the nonlinear amplitude-frequency characteristics of the test system quantitatively by theoretical method.

(3) We also build a biaxial theoretical model and consider the degrees of freedom in both directions, but it is very difficult to obtain analytical solutions theoretically and observe the amplitude-frequency characteristics from the equation. The part of theoretical analysis of biaxial model is explored as follows:

[Figure]

$$\frac{d}{dt}\left(\frac{\partial T}{\partial \dot{q}_j}\right) - \frac{\partial T}{\partial q_j} + \frac{\partial V}{\partial q_j} + \frac{\partial D}{\partial \dot{q}_j} = Q_j, j = 1,2,\cdots,n$$

Where: $T$- kinetic energy; $V$- potential energy; $D$- dissipated energy; $q_j$- generalized coordinate; $\dot{q}_j$ - generalized velocity; $Q_j$ - generalized force.

By selecting the generalized coordinate $q_1 = y_f$, $q_2 = y_e$. The displacement and velocity relationships of the test system can be obtained:

$$\begin{cases} R_f \cos\theta_f + L_f \sin\alpha_f = R_f + y_e \\ R_f \sin\theta_f + L_f\cos\alpha_f = L_f + y_f \\ -R_e \sin\theta_e + L_e\cos\alpha_e = L_e - y_e \\ R_e \cos\theta_e + L_e \sin\alpha_e = R_e + y_f \end{cases}$$

$$\begin{cases} -R_f\dot{\theta}_f \sin\theta_f + L_f\dot{\alpha}_f\cos\alpha_f = \dot{y}_e \\ R_f\dot{\theta}_f \cos\theta_f - L_f\dot{\alpha}_f \sin\alpha_f = \dot{y}_f \\ R_e\dot{\theta}_e \cos\theta_e + L_e\dot{\alpha}_e \sin\alpha_e = \dot{y}_e \\ -R_e\dot{\theta}_e \sin\theta_e + L_e\dot{\alpha}_e\cos\alpha_e = \dot{y}_f \end{cases}$$

$T$, $V$ and $D$ can be calculated as:

$$\begin{cases} T = \frac{1}{2}M(\dot{y}_f^2 + \dot{y}_e^2) + \frac{1}{2}m_f R_f^2 \dot{\theta}_f^2 + \frac{1}{2}m_e R_e^2 \dot{\theta}_e^2 \\ V = \frac{1}{2}k_f y_f^2 + \frac{1}{2}k_e y_e^2 \\ D = \frac{1}{2}c_f \dot{y}_f^2 + \frac{1}{2}c_e \dot{y}_e^2 \end{cases}$$

Some of the relevant terms in Eqs. (1) are obtained as:

$$\frac{\partial T}{\partial \dot{y}_f} = M\dot{y}_f + \left[\frac{m_f \cos^2\alpha_f}{\cos^2(\theta_f + \alpha_f)} + \frac{m_e \sin^2\alpha_e}{\cos^2(\theta_e - \alpha_e)}\right]\dot{y}_f + \left[\frac{m_f \cos\alpha_f \sin\alpha_f}{\cos^2(\theta_f + \alpha_f)} - \frac{m_e \cos\alpha_e \sin\alpha_e}{\cos^2(\theta_e - \alpha_e)}\right]\dot{y}_e$$

$$\frac{d}{dt}\left(\frac{\partial T}{\partial \dot{y}_f}\right) = \left[M + \frac{m_f \cos^2\alpha_f}{\cos^2(\theta_f + \alpha_f)} + \frac{m_e \sin^2\alpha_e}{\cos^2(\theta_e - \alpha_e)}\right]\ddot{y}_f + \left[\frac{m_f \cos\alpha_f \sin\alpha_f}{\cos^2(\theta_f + \alpha_f)} - \frac{m_e \cos\alpha_e \sin\alpha_e}{\cos^2(\theta_e - \alpha_e)}\right]\ddot{y}_e$$

$$+ \dot{y}_f \frac{d}{dt}\left[\frac{m_f \cos^2\alpha_f}{\cos^2(\theta_f + \alpha_f)} + \frac{m_e \sin^2\alpha_e}{\cos^2(\theta_e - \alpha_e)}\right]$$

$$+ \dot{y}_e \frac{d}{dt}\left[\frac{m_f \cos\alpha_f \sin\alpha_f}{\cos^2(\theta_f + \alpha_f)} - \frac{m_e \cos\alpha_e \sin\alpha_e}{\cos^2(\theta_e - \alpha_e)}\right]$$

**Comment 7:** The amplitude-frequency curves are incomplete with their peak points missing. From this figure, it can be observed that saddle node bifurcation occurs. Does the existence of saddle node bifurcation have effects on the results of biaxial fatigue test when the dynamic characteristics of such a system differ from those of the linear system?

**Response 7:**

Thank you very much for pointing out these problems, our response is as follows:

(1) The amplitude-frequency curves are incomplete with their peak points missing

The peak point is missing because the amplitude of the blade is different under different damping ratios. When the damping ratio is very small, the blade amplitude is very large, so the peak point is not displayed in the existing coordinate axis range. We will modify this issue in the revised paper and choose the damping ratio appropriately to ensure a complete curve.

(2) Effects of saddle node bifurcation on the results of biaxial fatigue test

Figure 5 shows the influence of different small parameters on the amplitude-frequency characteristics of the system. In fact, specific small parameter values mean specific working conditions, that is, when the virtual mass related parameters (such as $L$、$R$、$m$) are determined, the amplitude-frequency characteristics of the system will also be determined. Therefore, as long as the setups are determined, the dynamic characteristics of the test system will be determined, whether it is a single axis test or a biaxial test.

In addition, the amplitude hopping phenomenon, also known as dynamic bifurcation, also appears in Figure 5. In the simulation example, we can see that the resonance frequency changes in a relatively small range (The maximum variation of resonance frequency is about 2%), and the target amplitude of the tuning mass position in the fatigue test will not be large, so there will be no obvious dynamic bifurcation in fatigue test.

**Comment 8:** In Lines 200-202, authors state: "modal analysis is carried out and compared with the transfer-matrix method (TMM) and test data……" But there is no description about transfer-matrix method or the test. Please check.

**Response 8:**

Thank you very much for pointing out these problems, we will add the following to the revised paper in section 3.2.

Add content to the original lines 200-201: "To ensure the applicability of the model, modal analysis is carried out and compared with the transfer-matrix method (TMM) and the test data, taking the calculation of the flap-wise direction as an example, as shown in Table 1. The transfer matrix method is an approximate theoretical method used to calculate the natural frequencies and modes of systems with chain structures. The transfer matrix method separates the structure with inertia and elasticity and obtains the relationship between the discrete elements. The natural frequencies and modes of the systems can be solved according to the boundary conditions. The transfer matrix method belongs to the physical discrete method of continuous system, which is suitable for numerical solution of blade model. The three blades in Table 1 were all subjected to actual modal tests, and the obtained frequency data are obtained from the frequency domain analysis of actual test data. The actual blade modal test was carried out by hammer method."

We will make corresponding changes in the future revised paper and try our best to improve the manuscript. These changes will not influence the content and framework of the paper. Please do not hesitate to contact us if there are any questions. We appreciate for your hard work, and hope that the correction will meet with approval. Once again, thank you very much for your comments and suggestions.

Yours sincerely,
Jinlei Shi

---

## Author Comment (AC3)

Dear Reviewer #2:

Thank you for your comments and suggestions. Those comments are all valuable and very helpful for revising and improving our paper, as well as the important guiding significance to our researchers. We have studied comments carefully and have made correction which we hope meet with approval. Now I response the comments with a point by point. Full details of the files are listed. We sincerely hope that you find our response and modifications satisfactory and that the manuscript is now acceptable for publication.

Responds to the reviewer's comments:

**Comment 1:**

In line 200-202, authors should describe the transfer-matrix method (TMM) to let the reader better understand.

**Response 1:**

Thank you very much for pointing out these problems, we will add the following to the revised paper in section 3.2.

Add content to the original lines 200-201: "To ensure the applicability of the model, modal analysis is carried out and compared with the transfer-matrix method (TMM) and the test data, taking the calculation of the flap-wise direction as an example, as shown in Table 1. The transfer matrix method is an approximate theoretical method used to calculate the natural frequencies and modes of systems with chain structures. The transfer matrix method separates the structure with inertia and elasticity and obtains the relationship between the discrete elements. The natural frequencies and modes of the systems can be solved according to the boundary conditions. The transfer matrix method belongs to the physical discrete method of continuous system, which is suitable for numerical solution of blade model."

**Detailed description of TMM**

For the cantilever beam model of blades, a typical element is composed of massless beam and concentrated mass. The deflection $y$, angle $\theta$, bending moment $M$ and shear force $Q$ at each section are selected to form the state vector $Z$. The force analysis of massless beam and concentrated mass is shown in **Fig. m+1**, where $m_k$ is the concentrated mass, $l_k$ is the length of the beam, $EI_k$ is the bending stiffness of the beam, $x_k$ is the span-wise coordinate of the section $k$, the subscript $i$ represents the unit number, and the superscript L and R are used to distinguish the state vector of the left and right sections of the concentrated mass.

[Figure]

Fig. m Cantilever beam model of wind turbine blade

[Figure]

Fig. m+1 Force analysis diagram of massless beam and concentrated mass

The transfer matrix $H_i^B$ of the state vector for the massless beam from left to right is

$$
\begin{bmatrix} y \\ \theta \\ M \\ Q \end{bmatrix}_i^L = \begin{bmatrix} 1 & l_i & \dfrac{l_i^2}{2EI_i} & -\dfrac{l_i^3}{6EI_i} \\ 0 & 1 & \dfrac{l_i}{EI_i} & -\dfrac{l_i^2}{2EI_i} \\ 0 & 0 & 1 & -l_i \\ 0 & 0 & 0 & 1 \end{bmatrix} \begin{bmatrix} y \\ \theta \\ M \\ Q \end{bmatrix}_{i-1}^R \tag{n}
$$

The transfer matrix $H_i^M$ of the state vector for the concentrated mass from left to right is

$$
\begin{bmatrix} y \\ \theta \\ M \\ Q \end{bmatrix}_i^R = \begin{bmatrix} 1 & 0 & 0 & 0 \\ 0 & 1 & 0 & 0 \\ 0 & 0 & 1 & 0 \\ -m_i\omega^2 & 0 & 0 & 1 \end{bmatrix} \begin{bmatrix} y \\ \theta \\ M \\ Q \end{bmatrix}_i^L \tag{n+1}
$$

Since the state vector of the left section of the massless beam $i$ is the same as that of the right section of the concentrated mass $i$-1, and the mechanical state of the element is represented by the state vector of the right section of the concentrated mass, the transfer relationship of the state vector between adjacent elements is shown as $H_i = H_i^M H_i^B$. The transfer matrix of each element can be multiplied left to establish the total transfer matrix from the root to the tip as $H = H_n \cdots H_2 H_1$. Then, the transformation relationship between the root state vector $Z_0^R$ and the tip state vector $Z_n^R$ is shown as follows.

$$
\begin{bmatrix} y \\ \theta \\ M \\ Q \end{bmatrix}_n^R = \begin{bmatrix} h_{11} & h_{12} & h_{13} & h_{14} \\ h_{21} & h_{22} & h_{23} & h_{24} \\ h_{31} & h_{32} & h_{33} & h_{34} \\ h_{41} & h_{42} & h_{43} & h_{44} \end{bmatrix} \begin{bmatrix} y \\ \theta \\ M \\ Q \end{bmatrix}_0^R \tag{n+2}
$$

By substituting the boundary conditions at the root $y_0^R = 0$, $\theta_0^R = 0$ and the tip $M_n^R = 0$, $Q_n^R = 0$ into the **Eqs.** (n+2), the local matrix relationship is obtained as

$$
\begin{bmatrix} h_{33} & h_{34} \\ h_{43} & h_{44} \end{bmatrix} \begin{bmatrix} M_0^R \\ Q_0^R \end{bmatrix} = \begin{bmatrix} 0 \\ 0 \end{bmatrix}
$$

Since the deflection and shear force at the root are nonzero, according to the condition that the homogeneous linear equations have non-zero solutions, that is to say, the determinant of the coefficient matrix is zero, the following relation can be obtained

$$
h_{33}h_{44} - h_{34}h_{43} = 0 \tag{n+3}
$$

The coefficients of the total transfer matrix include the vibration frequency $\omega$, and the modal frequencies of the blade can be obtained by solving the above equation.

**Comment 2:**

In section 4.3, effects of virtual masses on biaxial test are considered and described in Figure 11. Authors should add a figure to describe the biaxial trajectory of the blade when the virtual masses are translational. The comparison of the two results (translation and rotation of virtual masses) can better illustrate the effect of virtual masses on blade biaxial test.

**Response 2:**

Taking 94m blade as an example, two conditions of virtual masses translation and rotation are compared to obtain the motion trajectory of the blade during biaxial test under the simulation environment. The exciting parameters and the tuning masses are the same as those in Section 3.3 and Table 3.

For rotation, select mechanism dimension as $R = 4m, L = 4m$.

[Figure]

Fig. m+2 Motion trajectory at exciting point

For ideal translation, select mechanism dimension as $R = \infty, L = \infty$ to simulate the virtual masses acting in only one vibration direction.

[Figure]

Fig. m+3 Motion trajectory at exciting point

When the exciting force amplitude keeps the same, the vibration amplitude is more stable and larger under the condition of virtual masses translation, and the motion trajectory is regular quadrilateral.

**Comment 3:**

What effect does the nonlinear effect introduced by virtual masses have on the actual test? Authors need to add further explanations.

**Response 3:**

As mentioned in the paper, the nonlinear effects during the blade fatigue testing are mainly due to the rotation of the virtual masses while the rotation radius and blade amplitude will affect the resonance frequency of the blade. According to the amplitude-frequency characteristic curve, when the excitation frequency deviates from the resonance region, the amplitude of the blade will drop sharply, resulting in the waste of energy of the fatigue test equipment. At the same time, specific areas of the blade are not sufficiently loaded to meet the certification requirements, but further loading requires additional energy consumption. Therefore, in view of the nonlinear effects introduced by virtual masses, it is necessary to improve the rapidity and accuracy of resonance frequency search during actual test.

**Comment 4:**

In Figure 8(d), there is (a) in this figure. Please check.

**Response 4:**

Thank you for your careful examination. The (a) in Fig. 8(d) is redundant and should be eliminated.

[Figure]

**Figure 8: (d) 94m blade in edge-wise direction**

We will make corresponding changes in the future revised paper and try our best to improve the manuscript. These changes will not influence the content and framework of the paper. Please do not hesitate to contact us if there are any questions. We appreciate for your hard work, and hope that the correction will meet with approval. Once again, thank you very much for your comments and suggestions.

Yours sincerely,
Jinlei Shi

---

## Author Comment (AC5)

Dear Reviewer #3:

Thank you very much for taking the precious time to read our paper and give valuable and helpful comments and suggestions. These suggestions are helpful to help us revise our paper better. We have studied comments carefully and have made correction which we hope meet with approval. Now I response the comments with a point by point. Full details of the files are listed. We sincerely hope that you find our response and modifications satisfactory and that the manuscript will be acceptable for publication.

Responds to the reviewer's comments:

**General comments:**

The results of this work are interesting and can be relevant to the progress of wind turbine rotor blade testing. However, the manuscript requires major review to be reconsidered for publication, as the key method used to derive the results is not sufficiently described and the connections between the different described methods are unclear. Overall, it is not clear which methods and theories are developed by the authors and which are considered from the literature. A clear description of origin and use within the work is mandatory. The connection between the presented analytical model (Chapter 2.1) the nonlinear vibration theory (Chapter 2.2) and the numerical simulation (Chapter 3) is unclear. It seems that the analytical differential equation (line 97 ff) could be transformed into the nonlinear vibration theory (line 140) but there is no description on how they are transformed and if this was done. The connection of the parameters in the nonlinear vibration theory to the investigated virtual mass system are unclear. Therefore, the applicability of the characteristics shown in Fig. 4 is questionable. Also, it is not stated how the simulation is connected to either of the analytical approaches of if any of the analytical results are used for the simulation. As the discussed results seem to be based only on the numerical simulation, the purpose of presenting the analytical methods is unclear. Additionally, the origin, functionality and modelling methods of the numerical simulation software, were not stated. The mentioned "numerical modal and harmonic analysis" (lines 187-188), which seem to be the key methods in this work, are generally linear simulation methods, and it is not explained how these were adapted to consider the described nonlinear characteristics and how the results were derived. Details on finite modelling of the test system were not given either. Also, it seems that a lot of model simplifications and assumptions were made compared to the behavior of real wind turbine rotor blades, but these assumptions are not clearly stated or explained (for details see comments 3-5 below). Furthermore, the presented results (chapter 4) for amplitude-frequency and radius-frequency are not put into context with realistic testing conditions. All these issues need to be addressed and the manuscript must be corrected accordingly. Further details and issues are described below.

**General responses:**

(1) In the revised manuscript, we will highlight our innovation points and clearly explain and cite the methods in the literature.

(2) We will further explain the connections between the chapters, and add corresponding text to explain the structure of the article to make it easier for readers to understand.

(3) We will explain the model in detail, explain the reasons for model simplification, and add corresponding simulation results.

**Major comments:**

**Comment 1 (lines 1-2):**

The title of the manuscript "Nonlinear Vibration Characteristics of Wind Turbine Blades Based on Virtual Mass Match" does not represent the presented work. Not the characteristics of the blade, but the characteristics of the virtual mass are examined. Also, the phrase "virtual mass match" is confusing as there is no "matching" presented in this work. Please change accordingly. (Proposed new title: "nonlinear vibration characteristics of virtual mass systems for wind turbine blade fatigue testing")

**Response 1:**

We agree with your suggestion. What we want to explore is the nonlinearity of virtual masses to the blade test system. we accept your proposed new title and will revise the title in the revised manuscript.

**Comment 2 (lines 7-20):**

The abstract only lists the separate steps of the method and a few results taken out of context. It is not clearly conveying the essence of the paper and it is missing an introduction into the topic and an interpretation of the result in the context of advancing the wind energy. Please correct accordingly.

**Response 2:**

Thanks for your suggestions, we will rewrite the abstract in the revised manuscript to explain the connections between the method and the results, and to illustrate the contribution of the paper to wind turbine blade fatigue test.

**Comment 3 (line 72):**

"The aim of virtual masses is to decouple the test load in the biaxial fatigue test." But most of the presented investigations are only concerning uniaxial blade testing. This suggests that the biaxial test can be derived by superimposing two uniaxial tests. As this assumption may be reasonable for linear behavior, when investigating nonlinear characteristics this assumption would not be valid. Please clarify the taken assumptions.

**Response 3:**

The nonlinear characteristic discussed in this paper refers to the influence of blade amplitude and seesaw radius on the resonant frequency of the test system. Although the purpose of virtual mass is to decouple the load in the biaxial loading, there will be inertial force coupling in the biaxial testing process, which will cause multiple factors to work together and make it difficult to analyze the system characteristics. Therefore, it is desirable to choose the uniaxial test to analyze the nonlinear influence introduced by the virtual masses, which does not mean that the biaxial test can be regarded as the linear superposition of the uniaxial test.

**Comment 4 (lines 72-74):**

"… inertial force that is transmitted to the blades through push rods, thereby adjusting the load distribution in the main vibration direction". This suggests that the blade displacement is assumed perfectly in line with the push rod. But as the blade is moving in biaxial testing the orientation of the push rod is constantly changing and not in line with either of the blades main directions (flap-wise, edge-wise). Please state and explain the taken assumptions.

**Response 4:**

We agree with you that the orientation of the push rod will change in the biaxial test, which will cause the direction of the inertial force to change. In fact, if the virtual mass mechanism is rotating, then the orientation of the push rod will change in either uniaxial test or biaxial test. This is also the main content of this paper.

Therefore, line 72 needs to be rewritten as: "The load distribution in the main vibration direction of the blade is adjusted by the component of the inertia force transmitted by the push rod in this direction. Because of the angle between the push rod direction and the vibration direction, blade displacement is not in line with the push rod.".

In addition, we need to further explain why the uniaxial model in the Chapter 2 was established. Because the analytical solution of the biaxial theoretical model is very complicated and it is difficult to assess the impact of single parameter on the system, we first set up a single-axis theoretical model to evaluate the effect of virtual masses rotation on the vibration characteristics of the test system, and then further analyze the effect of virtual masses on the biaxial test system through simulation software.

**Comment 5 (lines 76-77):**

"…frequency in edge-wise direction, the perturbation to the flap-wise direction is relatively small." This assumption is only valid if the edge-wise mode shape would be perfectly in line with the edge-wise blade direction. But typically, the first edge-wise mode shape of rotor blades also has a component in the flap-wise direction. Therefore, the push rod and virtual mass would need to be tilted accordingly to be perfectly in line with the motion, which would increase the effect on the flap-wise direction. Was this behavior considered in the presented work? Even, if the mode-shape is perfectly in line with the blade main direction, this assumption would only be valid for small angles beta. But the results in Chapter 4 suggest larger angles. Please clarify the used assumptions and state the chosen orientation of the push rod.

**Response 5:**

First of all, we agree with you very much. We do consider the differences between the model in our paper and the real blade model. We simplified the blade model in Chapter 2 mainly for the following three reasons:

Firstly, the uniaxial theoretical model is established to show more clearly the influence of the change of push rod direction (control variable method) caused by virtual masses rotation on the characteristics of the test system. On this basis, the simulation software is used to analyze the influence of the change of the push rod direction on the biaxial test.

Secondly, we did not have material, layup and other data to establish the finite element model of the blade, because the detailed blade model is the confidential file of the blade manufacturer. We can only get the stiffness and mass density of each section of the blade. That's why we didn't simulate

that edge-wise mode shape of rotor blades also has a component in the flap-wise direction.

Thirdly, we also further analyze the influence of the angle of the push rod on the flap-wise direction in the uniaxial model. There are two main factors in the disturbance of the flap-wise direction: one is the component force of the push rod in the flap-wise direction, and the other is the frequency of the component force. Because the component force of the push rod in the flap-wise direction is small, and according to vibration theory, when the frequency of the exciting force is far from the resonance frequency, the response (or amplitude) caused by the exciting force will be small.

To support this point, we take the following simulation.

Taking 84m and 94m blade as an example, $R = L = 4m$, the blade amplitude in edge-wise direction is about $Y = 1m$, the selected parameters as shown in section 4.2. From the following simulation results, the perturbation to the flap-wise direction is relatively small (0.5% of the edge-wise amplitude).

[Figure]

Figure: 84m blade amplitude of edge-wise direction

[Figure]

Figure: 84m blade perturbation amplitude of flap-wise direction

[Figure]

Figure: 94m blade amplitude of edge-wise direction

[Figure]

Figure: 94m blade perturbation amplitude of flap-wise direction

**Comment 6 (lines 82-92):**

The formulas are stated without any introduction, explanation or reference of origin. Also, no initial conditions are specified. Please correct accordingly.

**Response 6:**

Thank you for your suggestion, we will make corresponding changes in the revised manuscript.

**Comment 7 (line 106-108):**

The authors state "… it is difficult to obtain the corresponding analytical expression. Therefore, the numerical analysis methods are used to solve the equation." But it is not clearly explained which methods were used and how they were applied to the investigated system. Please clarify.

**Response 7:**

Thank you for pointing out our problems. We will make the following explanation in the revised manuscript.

Line 106: "… it is difficult to obtain the corresponding analytical expression. Therefore, the numerical analysis methods are used to solve the equation. A numerical simulation model based on the differential equation of the system motion is established in MATLAB SIMULINK, and the resonance frequency of the equivalent system is obtained after the initial displacement conditions are given. By modifying the value of the different parameter (m、k、R), the influence of the parameter change on the resonance frequency of the test system can be obtained."

**Comment 8 (line 112):**

"To illustrate this, numerical analysis is performed …" which numerical analysis was performed and how? Were the differential equations above used for this simulation?

**Response 8:**

Numerical analysis of differential equations is carried out in MATLAB SIMULINK to solve the transient response, see Response 7.

**Comment 9 (line 115 and Fig. 3):**

The authors state "the resonance frequency of the test system decreases nonlinearly" How were these resonance frequencies computed?

**Response 9:**

Please refer to Response 7.

**Comment 10 (lines 140-141):**

"$f(y) = 1 + \varepsilon\_1 y + \varepsilon\_2 y^2 + \varepsilon\_3\ y^3 + \varepsilon\_4 y^4$": Please explain how these formulas were derived for the investigated system, what they represent and add reference. What do the individual values for the small parameters represent and how can they be derived? If they cannot be derived, why is this relevant?

**Response 10:**

$$\left\{ M + m\frac{\cos^2\beta}{\cos^2(\theta-\beta)} \right\} \ddot{y} + c\dot{y} + ky + \frac{m\dot{y}^2\cos\beta}{\cos^4(\theta-\beta)}\left[\frac{\cos^2\beta\sin(\theta-\beta)}{R} - \frac{\sin^2\theta}{L}\right] = F(t) \tag{8}$$

According to Eqs. (8), the nonlinearity introduced by virtual masses rotation will be reflected in the inertia force term of the differential equation, which is marked by the red border. Since the $\theta$、$\beta$ in the inertial force term are all related to blade displacement,the characteristics of the test system are mainly affected by the inertial force provided by the virtual masses and the amplitude-frequency characteristic curve of the theoretical model cannot be obtained from Eqs. (8), Eqs. (9) is simplified on the basis of Eqs. (8). Small parameters in Eqs. (9) indicate the degree of nonlinearity.

According to the Theory of Nonlinear Vibration, the introduction of small parameters to solve the response of nonlinear systems by harmonic balance method is a basic method of nonlinear vibration theory. When the condition of the test system is determined, the value of the small parameter is also determined. That is, small parameters can be obtained by obtaining the vibration characteristics of a known system.

However, our paper borrows this approach to explain the nonlinearity brought about by virtual masses. We will add the corresponding references in the revised manuscript.

**Comment 11 (line 168):**

"the free vibration frequency of the nonlinear system with respect to the amplitude when there is no external excitation." How can there be vibration without external excitation, especially outside the natural frequency? Does this behavior apply to virtual mass systems? Please elaborate.

**Response 11:**

Line167-168: "However, the backbone of the support curve clusters is not straight but inclined. This backbone curve represents the variation of the free vibration frequency of the nonlinear system with respect to the amplitude when there is no external excitation." In a linear system, the free vibration frequency is called the natural frequency. They are determined by the characteristics of the vibration system itself, independent of amplitude and external excitation. In a nonlinear virtual masses system, the free vibration frequency is related to the amplitude of the blade, namely the backbone curve in line 167. The backbone curve (Eqs. (13)) is related to the characteristics of the nonlinear system and has nothing to do with the external excitation.

References:

Liu, Y. Z., Chen, L. Q.: Nonlinear vibrations. Higher Education Press, 2001.
非线性振动 – 百度学术 (baidu.com)

**Comment 12 (Fig. 5):**

The shown curves suggest that on the left side of the backbone for the same frequency and the same excitation there are three different states of vibration. How can this be possible? Does this behavior apply to the investigated virtual mass systems? If not, why is this relevant?

**Response 12:**

The same excitation in the curve corresponds to different vibration states, which is the frequency hopping phenomenon in nonlinear vibration. The excitation frequency of the blade-virtual masses test system only changes in a small range near the resonance frequency, so there is no hopping phenomenon. The frequency hopping phenomenon shown here is more obvious only when the blade displacement is very large. Moreover, when the influence of blade amplitude on the natural frequency of the system is discussed in the following paper, more attention is paid to the backbone curve in the shape of the black dotted line in the figure 5.

**Comment 13 (lines 186-188):**

What is the name and origin of the "blade motion simulation software" and how does it work? How can the "harmonic analysis" consider nonlinear characteristics, as harmonic analyses are generally linear simulations?

**Response 13:**

Thank you for pointing out our shortcomings and mistakes.

"blade motion simulation software" refers to "ADAMS" — multi-body dynamics simulation software. The model in this paper is based on the following steps: Firstly, the 3D model of the blade is established according to cross-section stiffness and mass density, and then Abaqus is introduced for mesh division and finite element model is generated. Then the finite element model is imported into ADAMS, and the push rod, seesaw, virtual masses and related motion pairs are established.

Here, the acquisition of resonance frequency should be based on the method of sweep frequency analysis rather than harmonic response analysis.

**Comment 14 (line 192):**

The authors state "The equivalent damping ratio of the blade changes during vibration, resulting in a change in the resonance frequency of the test system." How can the damping change if only the geometry of the system is different, and no damping elements are present? Is this structural damping? Please elaborate on the physics behind this.

**Response 14:**

We set the structural damping ratio in the simulation model, but did not consider the equivalent damping ratio obtained from air damping. This is because if the equivalent air damping ratio is considered, the vibration characteristics of the test system will also change, so that the nonlinear effects introduced by the virtual masses mechanism cannot be evaluated from the perspective of control variables.

In addition, we found in the actual blade test that the equivalent damping ratio obtained by the

attenuation method is positively correlated with the amplitude of the blade. According to vibration theory, the resonance frequency is the product of natural frequency and damping ratio correlation function, but this phenomenon is not the focus of this manuscript.

**Comment 15 (line 229):**

"sweep-frequency analysis is performed … to obtain the resonance frequencies…" What was the resolution of the frequency sweep? How was the resonance frequency identified? What are the vibrations amplitudes at different frequencies outside the resonance? Pleas add a plot like Fig. 5 for the behavior of the actual system.

**Response 15:**

The sweep frequency range is defined as a bandwidth of 0.02Hz near the first natural frequency in the flap-wise or edge-wise direction, with an action time of 1E4s and a resolution of 2e-6 Hz/sec. The frequency spectrum of the displacement of the exciting point of the blade under the sweeping excitation is analyzed, and the frequency corresponding to the peak point is the resonance frequency.

Taking the flapping direction loading of 94m blade as an example, the radius of the seesaw and the length of the push rod are set to 4m, the exciting force amplitude is 3500N, the sweep frequency range is 0.365~0.385Hz, the sweep frequency duration is 1E4s, and the maximum displacement value of the exciting point (59m) is 2m. The time-domain graph is shown as follows.

[Figure]

**Comment 16 (lines 262-263):**

The authors state "… effectively simulating additional masses that are directly fixed onto the blade." This is only a valid assumption if the displacement is exactly in line with the push rod (see comments 4 and 5). Please check.

**Response 16:**

Thank you for pointing out our problem. In fact, only when the length of the push rod is very long, the Angle of the push rod is basically unchanged during the fatigue test, and the displacement of the loading point of the blade is exactly the same as that of the push rod. At this time, the virtual masses can be regarded as directly fixed on the blade.

**Comment 17 (lines 271-273):**

the authors state "frequency sweep analysis are used to obtain the frequencies at which specific excitations are applied to the test system … The spatial coupling trajectory of the blade can be obtained…" How exactly were these frequencies and spatial trajectories derived? Was this done separately for flap-wise and edge-wise? If geometric nonlinearities of virtual masses are to be considered, the oscillations in flap-wise and edge-wise direction must not be evaluated separately as they influence each other. (see also comment 4) Please clarify.

**Response 17:**

Thank you again for pointing out our problem.

In the original manuscript, virtual masses are installed in both the flap-wise direction and the edge-wise direction and the resonance frequency of the test system is obtained by sweep frequency analysis of single input and single output in two directions respectively. Then, based on the results of sweep frequency analysis, resonant loading forces of corresponding frequencies are applied in both directions at the same time, and the spatial trajectory of blade motion is drawn after reaching a stable state. Because of the coupling effect of inertia force, the space trajectory appears irregular quadrilateral.

In fact, the oscillations in flap-wise and edge-wise direction must not be evaluated separately as they influence each other. Therefore, We will revise and more fully explain the simulation steps in the revised manuscript. The main ideas are as follows:

Firstly, the target amplitude of the blade is set, and then the resonance frequency of the blade in each direction is obtained by sweeping frequency iteration.

**Comment 18 (line 284-301):**

The conclusion provides an incomplete summary of the results which must be taken into context. Also, the conclusion is missing an interpretation of the results and an evaluation of usability for future wind energy technology. Please review.

**Response 18:**

Thanks for your suggestion, we will supplement and revise the conclusion in the revised manuscript.

**Detail comments:**

**Comment 19 (line 14):**

The authors state: "… the resonance frequency of the test system decreases by approximately 2%. …" Please clarify, which change in amplitude does this correspond to and what that means in the context of blade testing?

**Response 19:**

Taking a certain type of 80m+ blade as an example, when the flap-wise amplitude is 2.6m under uniaxial resonance loading, resonance frequency of the test system decreases by about 2%, so the inertial load of the blade decreases accordingly. In order to meet the test requirements, it is necessary to increase the exciting force amplitude. The increase of the amplitude of the blade will lead to the decrease of the resonance frequency, so the full load test needs to consume more energy.

**Comment 20 (lines 17-18):**

The authors state: "The decrease of the resonance frequency will also reduce the area of interest for blade load verification, …" How is the area of interest affected by the frequency decrease? Please clarify. The "area of interest" is usually part of the test requirements and cannot change due to test conditions as it defines the area that must be fully tested. Only the area which is actually fully tested can change, so please clarify definition.

**Response 20:**

Thank you for pointing out our problem. This sentence does not clearly express our point of view. The area of interest is defined by the blade test requirements and cannot change due to test conditions.

What we want to express is that at similar blade amplitudes, smaller resonance frequency means that the overall load of the blade will be reduced, which will make some areas fail to meet the target requirements. We will make corrections in the revised manuscript.

**Comment 21 (line 26):**

"… with over one million damage-equivalent loads cycles…" Even though, usually the cycles do not fall below 1 million it is not a requirement of the industry standards (IEC). Please review wording.

**Response 21:**

We agree with your suggestion and we will point out that this is not a standard requirement in the revised manuscript.

**Comment 22 (line 26):**

"… are performed at the 1st and 2nd natural frequency of the blade." It is required to test the blade in the flap-wise and edge-wise direction, which can be done utilizing the corresponding mode shapes. But this is not necessarily corresponding to the 1st and 2nd natural frequency. Please clarify.

**Response 22:**

Thank you for pointing out our problem. We will make corrections in the revised manuscript.

**Comment 23 (line 58):**

The author state "the modal characteristics of the testing system are basically determined, as shown in Fig. 1", but there is no process of determination shown in Fig. 1. Please clarify.

**Response 23:**

Line 57-58 will be revised as: "In the common fatigue test system, the additional masses are directly attached to the blade, as shown in Fig. 1 (a). When the additional masses are determined, the modal characteristics of the testing system are basically determined."

**Comment 24 (line 61-62):**

"…the inertia force generated by the virtual mass only acts in the direction of an individual blade mode." Please clarify which individual blade mode is concerned.

**Response 24:**

Here refers to the first mode of the edge-wise direction. We will make corrections in the revised manuscript.

**Comment 25 (line 71):**

Please explain the "Lagrange method", how it is applied and state a reference.

**Response 25:**

Lagrange equation is the fundamental equation of dynamics, we will add references about theoretical mechanics textbooks.

**Comment 26 (Fig. 2):**

The mass of the seesaw beam and the push rod were neglected in the modelling. Was this assumption investigated? Please state the assumption in the manuscript.

**Response 26:**

Because the work in this paper focuses on the nonlinear effect of the behavior of the virtual masses on the fatigue test of the blade, the mass of the push rod and the seesaw are ignored in modeling according to the control variable method, and only their geometric dimensions are considered. We will state the assumption in the revised manuscript.

**Comment 27 (lines 109-110):**

"… length of the push rod typically remains unchanged due to space limitations…" How is this "typical"? Is the later assumed length of 4m representative for all flap-wise, edge-wise and biaxial testing?

**Response 27:**

"typical" should be understood here as "generally", and the length of the push rod is always 4m in the following paper.

**Comment 28 (lines 115-126):**

These results seem as they are not generated from the Lagrange method and would therefore be in the wrong chapter. Please check.

**Response 28:**

These results are calculated by SIMULINK based on Lagrange dynamics equations, and the results are in the correct chapter.

**Comment 29 (line 117):**

why was the influence of k investigated but not the influence of M?

**Response 29:**

M represents the equivalent mass of the blade at the exciting point, and this paragraph investigates the influence of the virtual masses m. We will consider the influence of M in the revised manuscript.

**Comment 30 (Fig. 3):**

Using "/" as separator between measure and unit (e.g. "Y / m") is not advisable as it suggests division. Also, the measures in the legends are missing units (e.g. "m=1000" instead of "m=1000kg").

**Response 30:**

Thank you for your careful observation, we will make corresponding modifications according to your suggestions.

**Comment 31 (lines 124-126):**

How where the values for k and M derived and what do they represent?

**Response 31:**

k and M represent the equivalent stiffness and mass of the blade itself at the exciting point respectively and can be obtained by the principle of energy equivalence.

**Comment 32 (line 135):**

Please explain what "linear vibration theory" means in the presented context and add references.

**Response 32:**

Linear vibration theory refers to the study of the vibration processes of linear systems. We will add references in the revised manuscript.

**Comment 33 (line 138):**

"Thus, the weakly nonlinear dynamic equation…" Please clarify what "weakly" mean in this context?

**Response 33:**

"Weakly" means that although the vibration differential equation of the system is a nonlinear equation, compared with the differential equation of the linear system, it is only different in the acceleration term, and the influence of nonlinear factors is not particularly obvious according to the subsequent discussion.

**Comment 34 (line 142):**

Is the natural frequency $\omega\_n$ different from the resonance frequency? Please elaborate.

**Response 34:**

According to vibration theory, the resonance frequency is the product of natural frequency $\omega_n$ and damping ratio correlation function. In the use of linear systems, because the damping ratio is small, the resonance frequency is approximately considered as the natural frequency.

We will note the difference in the revised manuscript.

**Comment 35 (line 157):**

Please add references for "triangle transform" and "harmonic balance method"

**Response 35:**

We will add references for "triangle transform" and "harmonic balance method" in the revised manuscript.

**Comment 36 (line 166):**

"Similar to forced vibrations in linear systems, nonlinear systems also exhibit similar amplitude-frequency characteristic curves." What is the source of this information? Please add reference.

**Response 36:**

We will add references in the revised manuscript.

Liu, Y. Z., Chen, L. Q.: Nonlinear vibrations. Higher Education Press, 2001.

非线性振动 - 百度学术 (baidu.com)

**Comment 37 (line 169):**

"By setting B = 1 and $\zeta = 0\ldots$" What do these measures and the used values represent?

**Response 37:**

B is the coefficient and $\zeta$ represents the damping ratio. The above values are for the convenience of calculating the natural frequency of the undamped free vibration of the system.

**Comment 38 (lines 193-194):**

The authors state "In order to accurately assess the influence of virtual mass on the characteristics of the testing system, aerodynamic damping is not considered in the simulation model". But if the aerodynamic damping has an influence on the characteristics of the testing system it must be considered, as it is part of this system. Has it been confirmed that the aerodynamic damping does not influence the test system? Please elaborate.

**Response 38:**

Aerodynamic damping does have a nonlinear effect on the vibration characteristics of the test system, but it is not within the scope of this paper. We will explore the effect of aerodynamic damping on the test system in future work, and in fact we are already collecting relevant test data. In addition, the effect of virtual mass on the resonance frequency of the system cannot be reasonably evaluated by the introduction of aerodynamic damping, and it is difficult to model and analyze this problem due to the lack of blade airfoil.

**Comment 39 (Figure 6):**

what are the degrees of freedom of the test system? Can the blade move in any direction? How was the model discretized?

**Response 39:**

The installation methods of blade, virtual mass, push rod and seesaw have been described in line 62-64 and line 191. The freedom of the test system should be unlimited, because the blade is a flexible body. The 3D model of the blade is established according to cross-section stiffness and mass density. We will add relevant references to illustrate the rationality of the modeling.

**Comment 40 (line 200):**

What is the transfer-matrix method (TMM)? What is the relevance of comparing the TMM to the test besides the simulation?

**Response 40:**

The transfer matrix method is an approximate theoretical method used to calculate the natural frequencies and modes of systems with chain structures. The transfer matrix method separates the structure with inertia and elasticity and obtains the relationship between the discrete elements. The natural frequencies and modes of the systems can be solved according to the boundary conditions. The transfer matrix method belongs to the physical discrete method of continuous system, which is suitable for numerical solution of blade model. The three blades in Table 1 were all subjected to actual modal tests, and the obtained frequency data are obtained from the frequency domain analysis of actual test data. The actual blade modal test was carried out by hammer method.

The purpose of comparing TMM, test and simulation is to verify the rationality of the established blade model.

**Comment 41 (line 202):**

The authors state "… a high level of accuracy, with an error … less than 4%" But in the results a maximum of 2% deviation is visible which seems to be significant as they are repeated in the conclusion. Please comment on this discrepancy and correct accordingly.

**Response 41:**

4% refers to the relative error of natural frequency in blade simulation model and actual test, which is used to verify the accuracy of simulation model. 2% refers to the effect of blade amplitude on the resonant frequency of the system, describing the nonlinear effect of large amplitude on vibration characteristics. The two errors cannot be compared.

**Comment 42 (Table 1):**

Why was only the 1st modal frequency investigated? How much deviation does the 2nd modal frequency show? How is the deviation of the 102m blade relevant to the work as it is not investigated further? How were the test data derived? Please correct manuscript to answer these questions.

**Response 42:**

We will add contents in the revised manuscript.

The natural frequency of the 102m blade is obtained by the exact same actual test or modeling method, which is used here to verify the rationality of the modeling method. The three blades in Table 1 were all subjected to actual modal tests, and the obtained frequency data are obtained from the frequency domain analysis of actual test data. The actual blade modal test was carried out by hammer method.

**Comment 43 (line 207, Figure 7):**

"The values of the additional masses are shown in Table 2 and the Section properties of the blades are shown in Fig. 7" Why and how were the shown positions and magnitudes for the additional masses chosen? Are the flap stiffness, edge stiffness and density the only section properties considered in the simulation? What about torsional stiffness and coupling terms?

**Response 43:**

The position and values of the additional masses are provided by the blade designer. Generally, only the cross-section stiffness and linear density are considered in the simulation model, and the torsional natural frequency is much higher than the natural frequency in the direction of blade flapping and oscillation, so it is difficult to stimulate large torsional deformation.

**Comment 44 (line 226):**

"simulation results are fitted using relevant functions to verify the relationship" Which functions were used? How were they fitted? How well do they fit (coefficient of determination)? Was the relationship verified? Please elaborate and correct accordingly.

**Response 44:**

The fitting function is shown in equation (13) and (14), and we will add contents in the revised manuscript.

**Comment 45 (lines 232-235):**

Which position along the blade correspond to the stated amplitudes (blade tip or position of virtual mass)? Why are amplitudes of 2.6m for flap and 2.2m for edge significant? Are these amplitudes representative for real blade test of this size? Are the Amplitudes close to the geometrical limit where the push rod is parallel to the seesaw (beta - theta = 90°)? Would this be a realistic scenario where 2% frequency drop occurs? Please elaborate more on the interpretation of the results within the context of realistic blade testing.

**Response 45:**

The stated amplitude refers to the amplitude of the blade at the installation position of the virtual masses (i.e. the exciting point). The flap-wise amplitude of 2.6m and edge-wise amplitude of 2.2m are taken as examples to illustrate the decrease of the resonance frequency in combination with Figure 8, which does not mean that the situation will occur in real tests. Here, when the amplitude is close to 3m, the mechanism reaches the geometric limit of the push-rod parallel to the seesaw, so the limit requirements of the mechanism need to be considered. In the actual test, the blade generally requires amplitude of 2m for flap and amplitude of 1m for edge, so there will be no 2% frequency drop. Note that only the effect of the blade amplitude is discussed here, and the effect of the reduction of the seesaw radius is not considered.

We will elaborate more on the interpretation of the results within the context of realistic blade tests in the revised manuscript.

**Comment 46 (Fig. 8):**

The data seems to fit poorly to the fitted line. What can be causes for this? Especially in Fig. 8(c) the frequency seems to rise for low amplitudes before it drops? How can this be explained?

**Response 46:**

Due to the limitation of resonance frequency extraction precision in sweep frequency analysis, the fitting degree of data is affected. However, it is still acceptable on the whole. The deviation in the first half of Figure 8 (c) is due to the limited accuracy of sweep analysis in the case of small amplitude. Combined with the actual test requirements, we should pay more attention to the conditions of large amplitude in another half.

**Comment 47 (lines 248-249):**

Why are radii of 3m for flap and 2m for edge significant? Are these realistic for blade test of this size? Please correct as described in comment 45.

**Response 47:**

Here, the selection of the rotation radius of the seesaw considers the limitation of the amplitude of 2m for flap and the amplitude of 1m for edge (such amplitude is required for realistic testing) on the size of the mechanism. In order to reduce the occupied space, a smaller value is selected, so as to discuss the nonlinear influence of the rotation radius of the seesaw on the resonance frequency. The purpose of this study is different from that of comment 45.

**Comment 48 (Figure 10):**

How were these load distributions derived? Was constant force or constant displacement used for excitation? At what position along the blade was the excitation placed? What were the excitation frequencies? Please change manuscript to answer these questions.

**Response 48:**

In order to compare the influence of nonlinearity on the blade load distribution, the blade bending moment distribution can be calculated by using constant force excitation and inertial load provided by virtual mass motion. The installation position of the virtual masses is the same as the excitation position, and the specific values are shown in line 209 and Table 2. The excitation frequency is the resonant frequency of the respective vibration direction, which is obtained by the sweep frequency analysis (see Response 15 for details).

**Comment 49 (line 263):**

"As R decreases, the amplitude of blade loads reduces rapidly". The word "rapidly" seems inappropriate for a load amplitude drop of only 3% with a change in R from infinity to 3m. Please check.

**Response 49:**

Thank you for pointing out our poor choice of words and we will correct it.

**Comment 50 (lines 264-265):**

"…resulting in a reduction in the area of interest." See comment 20.

**Response 50:**

Thank you for pointing out our mistakes and we will correct it, please see response 20.

**Comment 51 (Figure 11):**

Please show, how would these trajectories change with different values for R?

**Response 51:**

We will add the corresponding conditions to analyze the trajectories change with different values for R.

**Comment 52 (line 288-289):**

"The square of the resonance frequency is inversely proportional to the polynomial steady-state response of the system." What does "polynomial steady-state response" mean? Which consequences can be taken from this? Please clarify.

**Response 52:**

This means that the square of the resonance frequency is inversely proportional to the polynomial of the steady-state amplitude of the system, as shown in Equation (13), which will determine the selection of the form of the fitting function in Figure 8.

**Comment 53 (line 290):**

"decreases by approximately 2%" Please state the reference from which it has decreased and the corresponding conditions of R=L=4m.

**Response 53:**

Here the decrease in resonance frequency is relative to the natural frequency obtained by modal analysis, as shown in Figure 8 (a). We will change manuscript to explain these questions.

**Comment 54 (line 294):**

"in the edge-wise direction, the radius of the seesaw has minimal impact on the resonance frequency" Why is 1.8% (flap-wise) significant enough to be mentioned but 1.0% (edge-wise) considered "minimal impact"? Also see comment 53. Please clarify.

**Response 54:**

Thank you for pointing out our poor choice of words and we will correct it.

**Comment 55 (lines 296-297):**

"… decreases by nearly 3% in the flap-wise direction under the given operating conditions" please clarify these operating conditions in terms of excitation force, displacement, and frequency.

**Response 55:**

The load distribution is shown in Figure 10, and the specific working conditions are shown in Response 48. We will also make changes in the revised manuscript.

**Comment 56 (lines 302-341):**

The reference are numbered in order of appearance but in the manuscript they are not referenced by number. Please either sort references by author of use numbers in text.

**Response 56:**

We saw the format requirements of the journal and also confirmed that the references appear in the same order in the text and appendix.

**Comment 57 (lines 305-306):**

The doi-link of reference [2] Liao et al. is not valid. Please check.

**Response 57:**

Thanks for pointing out the problem. The link of reference [2] will be changed as:
https://www.tynxb.org.cn/CN/Y2016/V37/I11/2785

**Comment 58 (lines 325, 327 and 337):**

There are three different references corresponding to "Melcher et al., 2020". Please clarify with refences are meant at the respective positions within the manuscript.

**Response 58:**

We will make changes in the revised manuscript. Thanks for your suggestions.

**Comment 59 (lines 332-334):**

The doi-link of reference [14] Zhang et al. is not valid. Please check.

**Response 59:**

Thanks for pointing out the problem. The doi-link of reference [14] will be changed as:
https://doi.org/10.1063/1.5112006

We will make corresponding changes in the future revised paper and try our best to improve the manuscript. These changes will not influence the content and framework of the paper. Please do not hesitate to contact us if there are any questions. We really appreciate for your hard work, and hope that the correction will meet with approval. Once again, thank you very much for your comments and suggestions.

Yours sincerely,
Jinlei Shi